# Benchmark data and model independent event classification for the large hadron collider

Thea Aarrestad[1], Melissa van Beekveld[2], Marcella Bona[3], Antonio Boveia[5], Sascha Caron[4], Joe Davies[3], Andrea De Simone[6,7], Caterina Doglioni[8], Javier M. Duarte[9], Amir Farbin[10], Honey Gupta[11], Luc Hendriks[4], Lukas Heinrich[1], James Howarth[12], Pratik Jawahar[13,1], Adil Jueid[14], Jessica Lastow[8], Adam Leinweber[15], Judita Mamuzic[16], Erzsébet Merényi[17], Alessandro Morandini[18], Polina Moskvitina[4], Clara Nellist[4], Jennifer Ngadiuba[19,20], Bryan Ostdiek[21,22], Maurizio Pierini[1], Baptiste Ravina[12], Roberto R. de Austri[16], Sezen Sekmen[23], Mary Touranakou[24,1], Marija Vaškevičiūte[12], Ricardo Vilalta[25], Jean-Roch Vlimant[20], Rob Verheyen[26], Martin White[15], Eric Wulff[8], Erik Wallin[8], Kinga A. Wozniak[27,1] and Zhongyi Zhang[4]

1 European Organization for Nuclear Research (CERN), 1211 Geneva 23, Switzerland
2 Rudolf Peierls Centre for Theoretical Physics, Oxford OX1 3PU, United Kingdom
3 Queen Mary University of London, UK
4 Nikhef, 1098 XG Amsterdam, Netherlands
5 Ohio State University, Columbus, OH 43210, USA
6 SISSA, 34136 Trieste TS, Italy
7 INFN, 34149 Trieste TS, Italy
8 Lund University, Lund, Sweden
9 University of California San Diego, La Jolla, CA 92093, USA
10 University of Texas at Arlington, Arlington, TX 76019, USA
11 Google Summer of Code
12 University of Glasgow, United Kingdom
13 Worcester Polytechnic Institute
14 Konkuk University, Seoul, Republic of Korea
15 University of Adelaide, Adelaide SA 5005, Australia
16 Instituto de Física Corpuscular, IFIC-UV/CSIC, 46980 Paterna, Valencia, Spain
17 Rice University, Houston, TX 77005, USA
18 RWTH Aachen University, 52062 Aachen, Germany
19 Fermi National Accelerator Laboratory, Batavia, IL 60510, USA
20 California Institute of Technology, Pasadena, CA 91125, USA
21 Harvard University, Cambridge, MA 02138, USA
22 The NSF AI Institute for Artificial Intelligence and Fundamental Interactions
23 Department of Physics, Kyungpook National University, Daegu, South Korea
24 National and Kapodistrian University of Athens, Athens 157 72, Greece
25 University of Houston, Houston, TX 77004, USA
26 University College London, London WC1E 6BT, United Kingdom
27 University of Vienna, 1010 Wien, Austria

## Abstract

We describe the outcome of a data challenge conducted as part of the Dark Machines (https://www.darkmachines.org) initiative and the Les Houches 2019 workshop on Physics at TeV colliders. The challenged aims to detect signals of new physics at the Large Hadron Collider (LHC) using unsupervised machine learning algorithms. First, we propose how an anomaly score could be implemented to define model-independent

signal regions in LHC searches. We define and describe a large benchmark dataset, consisting of $> 1$ billion simulated LHC events corresponding to $10\,\text{fb}^{-1}$ of proton-proton collisions at a center-of-mass energy of 13 TeV. We then review a wide range of anomaly detection and density estimation algorithms, developed in the context of the data challenge, and we measure their performance in a set of realistic analysis environments. We draw a number of useful conclusions that will aid the development of unsupervised new physics searches during the third run of the LHC, and provide our benchmark dataset for future studies at https://www.phenoMLdata.org. Code to reproduce the analysis is provided at https://github.com/bostdiek/DarkMachines-UnsupervisedChallenge.

Copyright T. Aarrestad *et al*.

This work is licensed under the Creative Commons
Attribution 4.0 International License.
Published by the SciPost Foundation.

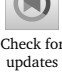

# Contents

# 1   Introduction and goals

**Why model-agnostic[1] searches are necessary:**   The Standard Model (SM) has been tremendously successful in describing a wide range of particle physics phenomena. Nevertheless, many questions still remain unanswered, e.g. the origin of neutrino masses, the nature of dark matter, and the dynamics of electroweak symmetry breaking. Therefore, it is commonly accepted that physics beyond the SM (BSM) is required and several theoretical arguments predict new particles at an energy scale that could be probed at the CERN Large Hadron Collider (LHC). A key requirement of the undertaking towards a new physics discovery is handling the huge amount of complex experimental data collected at the LHC. LHC data is analyzed for various experimental signatures, as predicted by specific SM extensions, generically referred to as "new physics models" or "beyond the SM" (BSM) models.

New physics models are tested using LHC data by optimizing data selection criteria on the energy, momenta and types of particle predicted by the model. Evidence for new particles typically manifests as an overproduction of events (compared to the SM)[2] in a specific data selection where the number of events expected from SM processes is compared to the number of measured events in statistical tests. Often the test is quantified with the help of a *p*-value, defined as the probability that a given result (or a more significant result) occurs under the SM hypothesis, in a frequentist statistical framework [1,2]. A typical requirement for the *discovery* of an *expected* signal (such as the Higgs particle) is $p < 3 \times 10^{-7}$ corresponding to 5 standard deviations ($5\sigma$) [3]. A hint of new physics requires that the "SM-only" hypothesis is highly disfavoured.

To date, no BSM physics has been discovered ($5\sigma$) at the LHC. However, the new physics could look different from the various hypotheses described above. This project deals with the question of how to search for a signal in collider data without adopting a specific signal hypothesis.

**Brief review of model-agnostic searches:**   A few attempts have been made to systematically search for new physics with minimal signal assumptions by scanning specific observables, such as the sum of the transverse momenta, or the invariant mass of particle decay products. Scans

---

   [1]Model-agnostic here means that we do not assume any specific extension of the Standard Model in the search strategy.

   [2]This could also be an underproduction in some models due to a negative quantum mechanical interference of the SM and new physics contributions. We do not consider this possibility in this study.

have been carried out with the help of model-agnostic (i.e. unsupervised) algorithms to locate anomalies. Such *general* searches without an explicit BSM signal assumption have been performed by the DØ Collaboration [4–7] at the Tevatron using an unsupervised multivariate signal detection algorithm termed SLEUTH, by the H1 Collaboration [8, 9] at HERA using a 1-dimensional signal detection algorithm that scans kinematic relevant quantities such as the (sum of) transverse momenta or the invariant mass, and by the CDF Collaboration [10, 11] at the Tevatron (using similar 1-dimensional algorithms). A version of these 1-dimensional signal detection algorithms that specifically searches for localized excesses ("bumps") and is used in general searches is known in the high energy physics (HEP) community as BUMPHUNTER [12]. At the LHC, searches for the presence of localized excesses have been performed by the ATLAS and CMS Collaboration in the dijet invariant mass distributions [13, 14]. Generic model-agnostic searches for new physics scanning thousands of analysis channels have also been performed at the LHC by ATLAS and CMS comparing data to SM simulations [15, 16]. In some of these analyses, the observation of one or more significant deviations in some phase-space region(s) can serve as a motivation to perform dedicated and model-dependent analyses where these "data-derived" phase-space region(s) can be used as signal region(s). Such a strategy can then determine the level of significance by testing the SM hypothesis in these signal regions in a second dataset (typically collected after the result of the model independent search). Since the signal region is known, a control region can also be defined to determine the background expectations in the signal region(s) (e.g. Ref. [15]).

One limitation of these model-agnostic approaches is the problem of multiple comparisons, or look-elsewhere effect [17], which reduces the significance of an observed deviation given that it may occur in any of the defined signal regions. Roughly, the $p$-value is reduced by a factor of the number of trials, i.e. the number of statistically independent signal regions that are considered. Thus, there is a fundamental trade-off between covering as many signatures as possible and maintaining good sensitivity to any individual deviation.

The field of machine learning (ML), sitting at the intersection of computational statistics, optimization, and artificial intelligence, has witnessed a significant step forward over the past decade. Research in ML has led to the development of new and enhanced anomaly detection methods that could be used and extended for applications employing LHC or astroparticle data. Examples of such outlier detection algorithms recently proposed for HEP include density-based methods [18], isolation forests, Gaussian mixture models [19], model-independent searches with multi-layer perceptrons [20], autoencoders [21–24], variational autoencoders [25–28], adversarially trained networks [29], ML extended bump-hunting algorithms [30–38], and self-supervision [39].

The recent LHC Olympics (LHCO) [40] studied anomaly detection techniques in three different black boxes. Using unsupervised, weakly supervised, and semi-supervised approaches, the contestants were tasked with determining if a black box contained new physics, and if so, to identify its properties. In contrast to this paper, which deals with a comparison of unsupervised event classification with final reconstructed detector objects, the LHCO task was to search for a (possible) signal as an overdensity in the data and thus to determine the signal and evaluate the background. The LHCO data consist of the reconstructed particles prior to the final high-level reconstruction of objects. Events could consist of up to 700 particles and jet clustering had to be used, while events in this article consist of up to 20 fully reconstructed particles. A few methods were able to detect the resonance in Box 1, but none were successful for Box 3. Meanwhile, Box 2 included only SM events, yet multiple algorithms claimed detection of a high-mass resonance. The outcome of this exercise highlights the need for more dedicated studies of anomaly detection techniques as well as publicly available data to develop and compare methods across common benchmarks.

Searches for new physics at the LHC typically define subsets of data potentially enriched

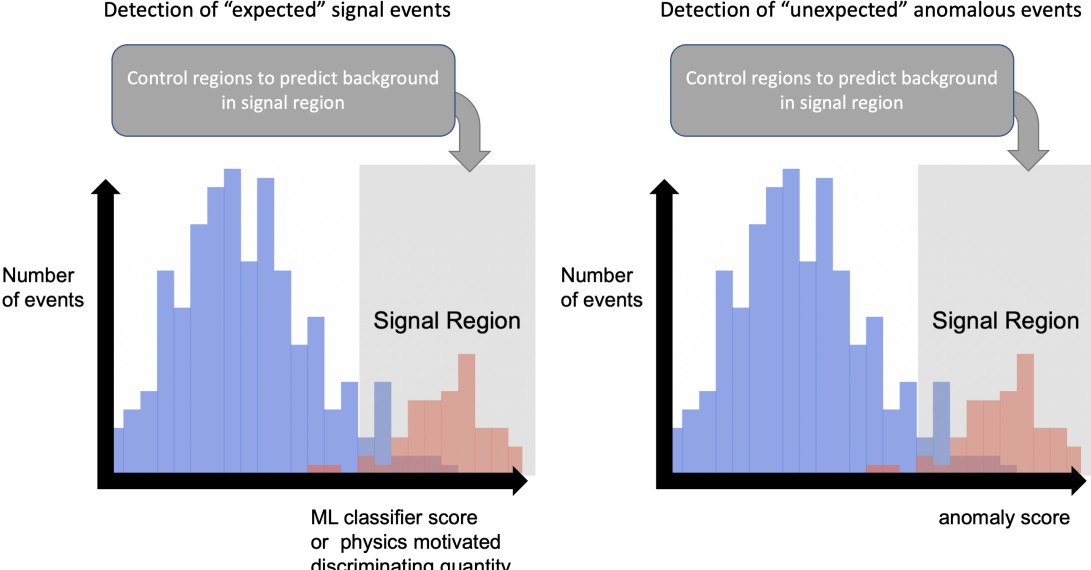

Figure 1: Illustration of the strategies for defining signal regions with different traditional approaches (left picture) and the anomaly score proposed and examined in this article (right picture). The same histogram is used to emphasize the similarity of the strategies, however, we do not expect the strategies to yield the exact same results.

with signal ("signal region" or "search region") with a series of signal selection criteria. When searching for new resonances, criteria are typically placed on the angular distance between the decay products, on the invariant mass of the particles in a given final state (e.g., dijet or dilepton) or the missing transverse momentum. In the context of SUSY, other examples of such variables include the effective mass [41], the transverse mass [1], $\alpha_T$ [42], razor [43], event shape variables like sphericity [44], and the recursive jigsaw technique [45]. These criteria are typically designed and set to optimize the selection of a predefined set of events from an assumed BSM process. Other approaches to define a BSM signal region involve selection criteria on a supervised machine learning classifier trained to distinguish a potential signal from background events.

In addition to the signal regions, there are also so-called "control regions". Control regions are defined such that they have an increased contribution from background processes. These control regions are then used to determine the expectation of background processes in the signal region.

**Approach in this paper:**   Here we propose a different approach with a rather small but important change in the search strategy. Our study aims to define an **anomaly score** signal range by imposing a lower threshold on the anomaly score defined by an unsupervised algorithm (right side of Fig. 1) event-by-event. These algorithms are trained on simulated SM events only; i.e. without defining a signal. The algorithms can also be trained directly with real, unlabeled collision data under the assumption that the signal is rare compared to the background, which would make the method more robust to detector noise, although this is not done in this paper. For each event the anomaly score is defined such that events that look different or are unlikely to be generated from SM processes will have high anomaly scores.

As shown in Fig. 1, events with high anomaly scores might accumulate in the signal region. In the signal region, a statistical test between data and SM expectation is then used to

determine whether the data contains an excess of abnormal events.

It is important to emphasize that this way of defining a signal region is a rather small modification of existing search strategies. Therefore, most of the methods for determining the expectation of backgrounds in the signal regions and methods for performing the statistical test can be retained from existing analyzes. In addition, "anomaly score" signal ranges can be incorporated into existing searches with relative ease.

We would like to stress that, being unsupervised, i.e., model independent, this approach cannot be optimized according to a single well defined criterion. The optimal objective is typically to find events that are out of the phase space distribution of SM events. The question that remains is how such an anomaly score can optimally be defined and how well it works.

**Look elsewhere effect:** In contrast to model agnostic searches which scan over thousands of analysis channels, we emphasize that anomaly detection does not require this. For instance, using any one of the methods of this paper, it would be possible to define a signal region based on the anomaly score (e.g. a cut reducing the background by a factor of 100). As in other searches, a control region can also be defined. Using the control region, the background in the signal region can be predicted with a background-only fit. Finally, a p-value or significance can be obtained with the data and background prediction in the signal region. In this manner, there are only as many trial factors as there are signal regions (one would not use all of the methods studied in this paper simultaneously).

**Benchmark datasets and Dark Machines:** While this document does not address how detector-related anomalies contribute to the anomaly score (and, therefore, whether our results on simulation are representative of what would happen in data), it does describe and compare the performance of a number of possible algorithms and anomaly scores resulting from a data challenge carried out in the "unsupervised searches at colliders" group of the Dark Machines initiative. Dark Machines is an open research collective of physicists, statisticians and data scientists. Dark Machines aims to answer questions about the dark universe and new physics using the most advanced techniques that data science provides us with. In particular, Dark Machines organizes data challenges for problems in (astro) particle physics [46, 47].

Large datasets of > 1 billion simulated events that were created for these challenges are described and made available (see Tab. 4). This dataset will remain useful for various future applications. Additionally, a "secret" (i.e., unlabeled) dataset is provided that can be used to benchmark the anomaly scores of future ML algorithms[3].

## 2 Description of the challenge and the dataset

The objective for the challenge is to give an event-by-event classification between SM events (background) and events produced by BSM processes (signal). For this challenge, it is assumed that these "signal events" look different from the background or have a significantly lower probability of occurrence. The output of the classification algorithms is an "anomaly score" for each event. This is a continuous number between 0 (for background) and 1 (for signal). The determination of the anomaly score depends on the employed algorithm (see section 3). In what follows, we describe the generation of the data that is used for the challenge.

---

[3]Readers interested to run their own algorithms on the secret dataset are invited to submit their request to the contact persons of this document to arrange such a performance assessment.

Table 1: Definition of symbols used for final-state objects. Only $b$-quark jets are tagged, no $\tau$- or $c$-jets have been defined.

| Symbol ID | Object |
|:---:|:---:|
| j | jet |
| b | $b$-jet |
| e– | electron ($e^-$) |
| e+ | positron ($e^+$) |
| m– | muon ($\mu^-$) |
| m+ | antimuon ($\mu^+$) |
| g | photon ($\gamma$) |

## 2.1 Data generation procedures

We simulate proton-proton collision events similar to those occurring at the LHC at a center-of-mass energy of 13 TeV. The generation of events for the signal and the background processes has been performed at leading order (LO) with up to two extra partons in the final state using `MG5_aMC@NLO` [48]. The choice of the unphysical renormalisation and factorisation scales was set dynamically to be equal to the transverse mass of the $2 \to 2$ system resulting from a $k_{\mathrm{T}}$ clustering. The convolution of the parton-level matrix elements with non-perturbative parton distribution functions (PDFs) was performed using LHAPDF6 [49] where we use the `NNPDF31_lo` PDF set with $\alpha_s(M_Z^2) = 0.118$ [50] assuming the 5 flavor number scheme of the proton[4]. To add parton showering to the parton-level generated samples, we interface `MadGraph` to `Pythia` version 8.239 [51]. The matching of the matrix elements with different parton multiplicities to the parton shower algorithm was performed using the MLM merging scheme [52] and a merging scale of $Q_0 = 30$ GeV. In the process of event generation, we did not simulate the effects of multiple parton interactions within the same or neighboring bunch crossings (pileup). The corresponding signal and background cross sections were not reweighted to the higher-order and/or resummed cross sections which exist in the literature. A fast detector simulation was performed using `Delphes` version 3.4.2 [53] using a modified version of the ATLAS detector card[5]. In the process of the detector simulation, we used `FastJet` [54] to perform jet clustering with the anti-$k_t$ algorithm and a jet radius of $R = 0.4$. Jets coming from $b$-quarks are tagged in the `Delphes` card similar to [55]. A repository of the data scripts that are used to generate the events can be found in [56].

The final-state physics objects as described in Tab. 1 are stored in a one-line-per-event CSV text file (see section 2.2 for details). A collision event results in a variable number of objects. An event is stored when at least one of the following requirements is fulfilled[6]:

- At least one jet or a $b$-jet with transverse momentum $p_{\mathrm{T}} > 60$ GeV and pseudorapidity $|\eta| < 2.8$, or

- at least one electron with $p_{\mathrm{T}} > 25$ GeV and $|\eta| < 2.47$, except for $1.37 < |\eta| < 1.52$, or

---

[4]This results in a large cross section for the $W^+W^-$ production ($\sigma_{WWjj} = 244$ pb, whereas one would find $\sigma_{WWjj} = 82.1$ pb in the 4 flavor number scheme). The reason for the large differences between the two values is that single-resonant and double-resonant top quark mediated diagrams contribute to the one-parton and two-parton exclusive samples to the merged cross section in the 5-flavor scheme.

[5]The `Delphes` card is available at https://github.com/melli1992/unsupervised_darkmachines/blob/master/delphes_card_ATLAS.dat. See the card for information about object isolation.

[6]We use a Cartesian coordinate system with the $z$ axis oriented along the beam axis, the $x$ axis on the horizontal plane, and the $y$ axis oriented upward. The $x$ and $y$ axes define the transverse plane, while the $z$ axis identifies the longitudinal direction. The azimuth angle $\phi$ is computed with respect to the $x$ axis. The polar angle $\theta$ is used to compute the pseudorapidity $\eta = -\log(\tan(\theta/2))$. The transverse momentum ($p_{\mathrm{T}}$) is the projection of the particle momentum on the $(x, y)$ plane. We fix units such that $c = \hbar = 1$.

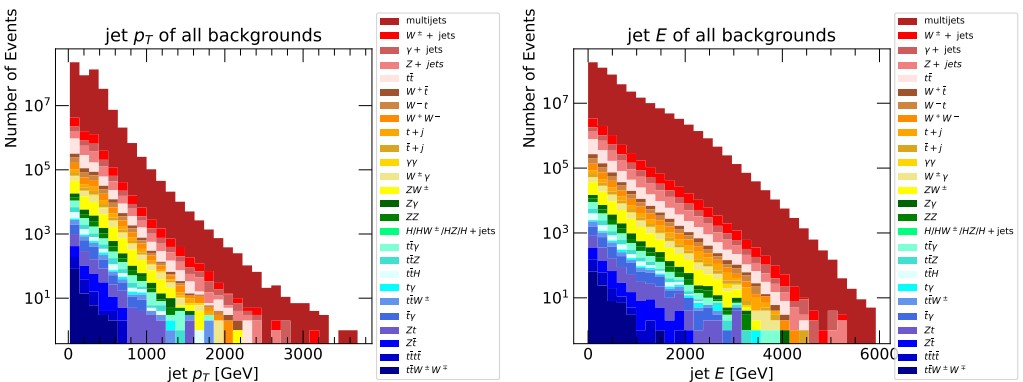

Figure 2: Transverse momentum $p_T$ (left) and energy $E$ (right) in GeV of the jets for all backgrounds.

- at least one muon with $p_T > 25$ GeV and $|\eta| < 2.7$, or

- at least one photon with $p_T > 25$ GeV and $|\eta| < 2.37$.

Of course, these are unrealistic requirements in terms of what an experiment can afford to record after the online data selection (trigger) system, but our aim is to create a flexible dataset that allows for different types of studies that might require different selection criteria. The $\eta$-restriction on the electrons models a veto in the crack regions as often applied in LHC analyses. Such a veto can also be applied to photons by the user. For the SM processes with the largest cross sections ($W^{\pm}/\gamma/Z$ + jets and QCD jet production) we have additionally applied requirements on $H_T > 100$ GeV and 600 GeV respectively to make the data generation manageable. The observable $H_T$ is defined as the scalar sum of the transverse momenta of all jets (with $p_{T,j_i} > 20$ GeV and $|\eta_{j_i}| < 2.8$):

$$H_T = \sum_i |p_{T,j_i}|. \tag{1}$$

Therefore, if one includes any of these processes in an analysis, one must make sure that the same requirements are also applied to the other processes[7], which impacts the production cross sections (and therefore the event weights) that are indicated in Tab. 2[8].

The requirements on the final state objects that are stored in the text files are:

- jet or $b$-jet: $p_T > 20$ GeV and $|\eta| < 2.8$,

- electron/muon: $p_T > 15$ GeV and $|\eta| < 2.7$,

- photon: $p_T > 20$ GeV and $|\eta| < 2.37$.

This means that, for example, a jet with $p_T = 10$ GeV is not included in the dataset. The detector simulation as performed by Delphes removes any electrons with $|\eta| > 2.5$, as the reconstruction efficiency is set to 0 beyond that point.

All relevant SM (background) processes that have been generated are summarized in Tab. 2. For each process, the total number of generated events ($N_{\text{tot}}$) is at least the number that is needed for 10 fb$^{-1}$-equivalent of data ($N_{10\,\text{fb}^{-1}}$). In Figs. 2-5 we show the (stacked) distributions of the kinematic variables $E$, $p_T$, $\eta$, and $\phi$ of the jets and leptons in all of the generated background processes. In Fig. 6 we show the number of jets, $N_{\text{jet}}$, and leptons, $N_{\text{lepton}}$,

---

[7]In general, the same selection must be applied to all samples within a given analysis.
[8]Note that for this challenge the event weights are not used.

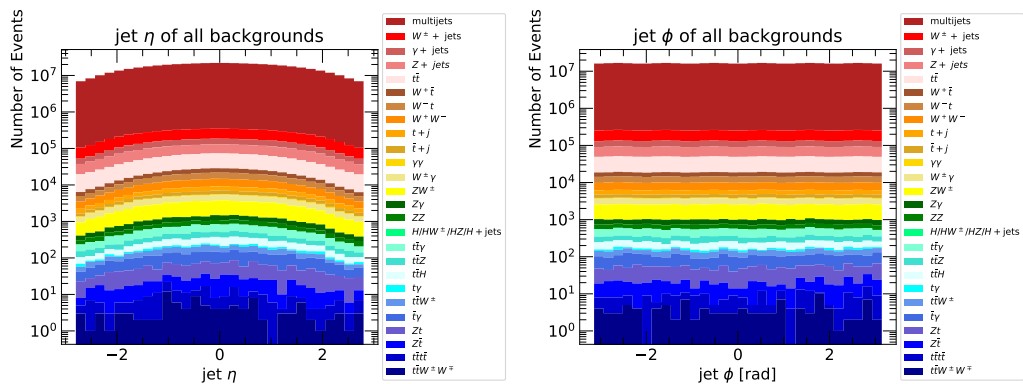

Figure 3: Pseudorapidity $\eta$ (left) and azimuthal angle $\phi$ (right) of the jets for all backgrounds.

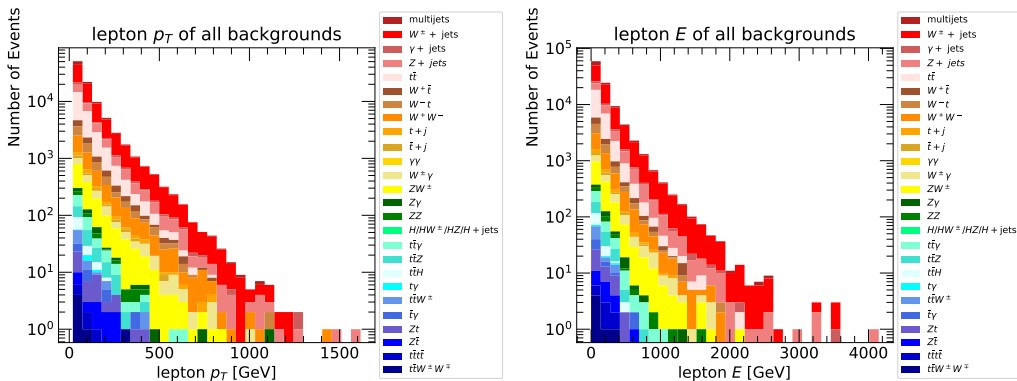

Figure 4: Transverse momentum $p_\mathrm{T}$ (left) and energy $E$ (right) in GeV of the leptons ($e^+$, $e^-$, $\mu^+$, $\mu^-$) for all backgrounds.

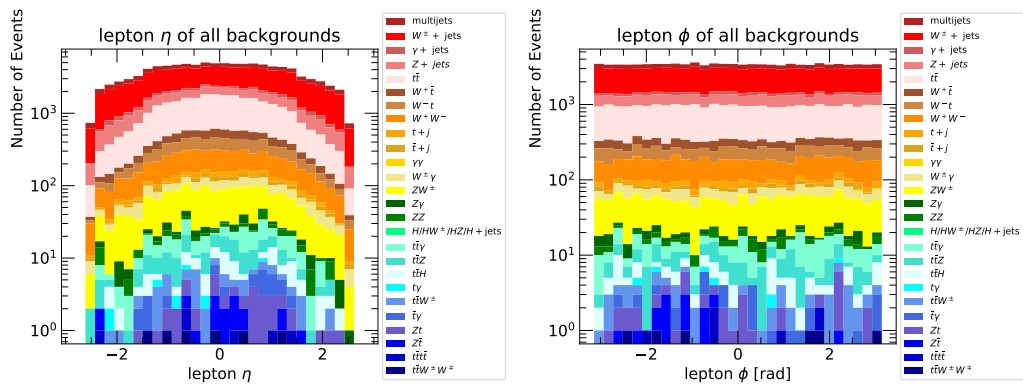

Figure 5: Pseudorapidity $\eta$ (left) and azimuthal angle $\phi$ (right) of the leptons ($e^+$, $e^-$, $\mu^+$, $\mu^-$) for all backgrounds.

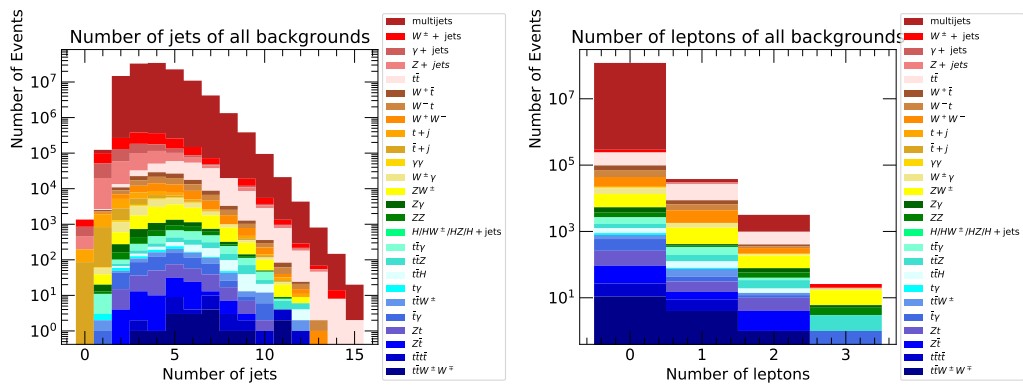

Figure 6: Number of jets (left) and leptons (right).

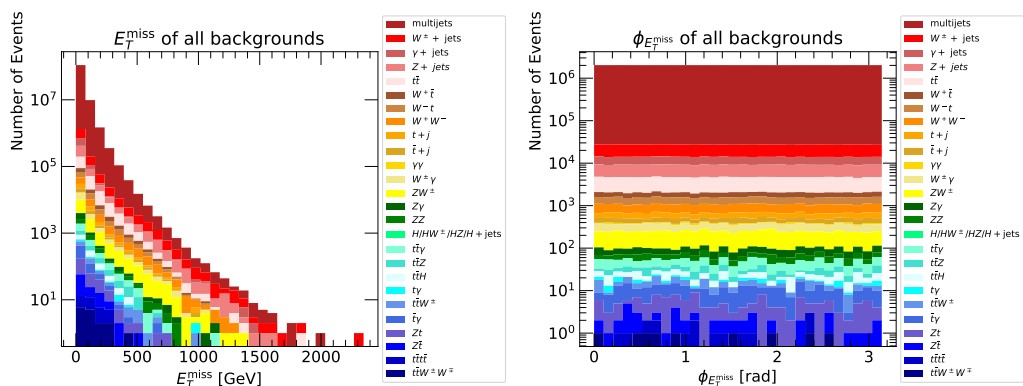

Figure 7: Missing transverse energy $E_T^{\mathrm{miss}}$ in GeV and azimuthal angle $\phi_{E_T^{\mathrm{miss}}}$ for all backgrounds.

Table 2: Generated background processes (first column) with the corresponding identification (second column), the LO cross section $\sigma$ in pb (third column) and the total number of generated events $N_{\text{tot}}$ (fourth column). In the last column, we also indicate the number of events corresponding to 10 fb$^{-1}$ of data ($N_{10\,\text{fb}^{-1}}$).

| SM processes | | | |
|---|---|---|---|
| Physics process | Process ID | $\sigma$ (pb) | $N_{\text{tot}}$ ($N_{10\,\text{fb}^{-1}}$) |
| $pp \to jj(+2j)$ | njets | $19718_{H_{\text{T}}>600\text{GeV}}$ | 415331302 (197179140) |
| $pp \to l^{\pm}\nu_l(+2j)$ | w_jets | $10537_{H_{\text{T}}>100\text{GeV}}$ | 135692164 (105366237) |
| $pp \to \gamma j(+2j)$ | gam_jets | $7927_{H_{\text{T}}>100\text{GeV}}$ | 123709226 (79268824) |
| $pp \to l^+l^-(+2j)$ | z_jets | $3753_{H_{\text{T}}>100\text{GeV}}$ | 60076409 (37529592) |
| $pp \to t\bar{t}(+2j)$ | ttbar | 541 | 13590811 (5412187) |
| $pp \to t+\text{jets}(+2j)$ | single_top | 130 | 7223883 (1297142) |
| $pp \to \bar{t}+\text{jets}(+2j)$ | single_topbar | 112 | 7179922 (1116396) |
| $pp \to W^+W^-(+2j)$ | ww | 82.1 | 17740278 (821354) |
| $pp \to W^{\pm}t(+2j)$ | wtop | 57.8 | 5252172 (577541) |
| $pp \to W^{\pm}\bar{t}(+2j)$ | wtopbar | 57.8 | 4723206 (577541) |
| $pp \to \gamma\gamma(+2j)$ | 2gam | 47.1 | 17464818 (470656) |
| $pp \to W^{\pm}\gamma(+2j)$ | Wgam | 45.1 | 18633683 (450672) |
| $pp \to ZW^{\pm}(+2j)$ | zw | 31.6 | 13847321 (315781) |
| $pp \to Z\gamma(+2j)$ | Zgam | 29.9 | 15909980 (299439) |
| $pp \to ZZ(+2j)$ | zz | 9.91 | 7118820 (99092) |
| $pp \to h(+2j)$ | single_higgs | 1.94 | 2596158 (19383) |
| $pp \to t\bar{t}\gamma(+2j)$ | ttbarGam | 1.55 | 95217 (15471) |
| $pp \to t\bar{t}Z$ | ttbarZ | 0.59 | 300000 (5874) |
| $pp \to t\bar{t}h(+1j)$ | ttbarHiggs | 0.46 | 200476 (4568) |
| $pp \to \gamma t(+2j)$ | atop | 0.39 | 2776166 (3947) |
| $pp \to t\bar{t}W^{\pm}$ | ttbarW | 0.35 | 279365 (3495) |
| $pp \to \gamma\bar{t}(+2j)$ | atopbar | 0.27 | 4770857 (2707) |
| $pp \to Zt(+2j)$ | ztop | 0.26 | 3213475 (2554) |
| $pp \to Z\bar{t}(+2j)$ | ztopbar | 0.15 | 2741276 (1524) |
| $pp \to t\bar{t}t\bar{t}$ | 4top | 0.0097 | 399999 (96) |
| $pp \to t\bar{t}W^+W^-$ | ttbarWW | 0.0085 | 150000 (85) |

for the generated backgrounds. The $E_{\text{T}}^{\text{miss}}$ and $\phi_{E_{\text{T}}^{\text{miss}}}$ distributions are shown in Fig. 7, and the $H_{\text{T}}$ distribution is shown in Fig. 8. Note that, only for Fig. 8, we have filtered out the events with $H_{\text{T}} < 600$ GeV. For the other figures, we show the events for all values of $H_{\text{T}}$ for most backgrounds, except for the ones with tags njets ($H_{\text{T}} > 600$ GeV), w_jets, gam_jets and z_jets ($H_{\text{T}} > 100$ GeV).

For the BSM scenarios (signal events) we have chosen a selection of SUSY and non-SUSY BSM processes. While these models do not cover the range of possible BSM signatures, they are motivated by having a dark matter particle which escapes the detector.

- The $Z' + $ monojet [57–59] contains a 2 TeV $Z'$ which decays fully invisibly to 50 GeV Dirac dark matter. This process is denoted as monojet_Zp2000.0_DM_50.0 in the figures.

- The $Z' + W/Z$ [57–59] also contains a 2 TeV $Z'$ which decays fully invisibly to 50 GeV Dirac dark matter. This process is denoted as monoV_Zp2000.0_DM_50.0 in the figures.

- The $Z' + $ single top process [57–59] involves a 200 GeV $Z'$. This process is denoted as monotop_200_A in the figures.

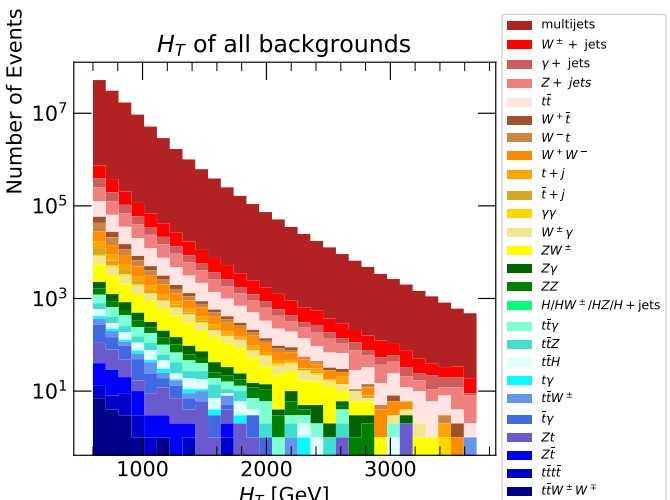

Figure 8: The scalar sum of the jet transverse momenta $H_\mathrm{T}$ in GeV (see Eq. (1)) for all backgrounds, with $H_\mathrm{T} > 600$ GeV imposed.

- The $Z'$ in lepton-violating $U(1)_{L_\mu - L_\tau}$ [60, 61] process involves a 50 GeV $Z'$ which decays to leptons and neutrinos. There are two processes included, `pp23mt_50` has three leptons in the final state and `pp24mt_50` has four leptons in the final state.

- The R-parity violating (RPV) SUSY [62, 63] stop-stop process has pair production of 1 TeV supersymmetric stops which decay to leptons and $b$-quarks. This process is denoted as `stlp_st1000` in the figures.

- The RPV SUSY [62, 63] squark-squark process has 1.4 TeV squark pair production. The neutralino has a mass of 800 GeV. The squarks decay down to jets. This process is denoted as `sqsq1_sq1400_neut800` in the figures.

- The SUSY [64–66] gluino-gluino process involves the pair production of gluinos which eventually decay to jets and neutralinos (missing energy). We include two different benchmark sparticle mass spectra. In the first, the gluinos have a mass of 1.4 TeV and the neutralinos have a mass of 1.1 TeV. This is denoted as `glgl1400_neutralino1100` in the figures. In the second spectrum, the gluinos have a mass of 1.6 TeV and the neutralinos have a mass of 800 GeV. This is denoted as `glgl1600_neutralino800` in the figures.

- The SUSY [64–66] stop-stop process has pair produced stops which decay to a top quark and a neutralino (missing energy). The stops have a mass of 1 TeV and the neutralinos have a mass of 300 GeV. This is denoted as `stop2b1000_neutralino300` in the figures.

- The SUSY [64–66] squark-squark process contains 1.8 TeV squarks which decay to jets and neutralinos (missing energy). The mass of the neutralinos is 800 GeV. This is denoted as `sqsq_sq1800_neut800` in the figures.

- The SUSY [64–66] chargino-neutralino processes involve the charged-current production of a chargino and neutralino. The chargino decays to a $W$ and a neutralino. There are two mass spectra considered. The first has a 200 GeV chargino and a 50 GeV neutralino, denoted as `chaneut_cha200_neut50` in the figures. The second contains a 250 GeV chargino and a 150 GeV neutralino and is denoted as `chaneut_cha250_neut150`.

- The SUSY [64–66] chargino-chargino process is the neutral current pair production of charginos which decay to a $W$ and neutralino. There are three considered mass spectra. The first is denoted as `chacha_cha300_neut140` and contains 300 GeV charginos and 140 GeV neutralinos. The second is more split, with the charginos at 400 GeV and the neutralinos at 60 GeV and is denoted as `chacha_cha400_neut60`. The final spectrum is heavier with 600 GeV charginos and 200 GeV neutralinos. This is denoted as `chacha_cha600_neut200`.

These scenarios are summarized in Tab. 3.

We then divide the background and signal events into separate (non-orthogonal) signal regions, referred to as channels. Channel 1 looks for hadronic activity with lots of missing energy. This is good for mono-jet type signatures of dark matter as well as any of the colored SUSY signals. Both of the channel 2 options reduce the background by requiring leptons, which then are more sensitive to signals which have an electroweak charge (such as the charginos and neutralinos). Channel 3 is targeted to be more inclusive and catches most of the signals except for the softer electroweak signals. The channels are defined as follows:

- Channel 1 (214 000 SM events):

$$H_{\mathrm{T}} \geq 600 \text{ GeV}, \quad E_{\mathrm{T}}^{\mathrm{miss}} \geq 200 \text{ GeV}, \quad E_{\mathrm{T}}^{\mathrm{miss}}/H_{\mathrm{T}} \geq 0.2, \tag{2}$$

with at least four (b)-jets with $p_{\mathrm{T}} > 50$ GeV, and one (b)-jet with $p_{\mathrm{T}} > 200$ GeV.

- Channel 2a (20 000 SM events):

$$E_{\mathrm{T}}^{\mathrm{miss}} \geq 50 \text{ GeV}, \tag{3}$$

and at least 3 muons/electrons with $p_{\mathrm{T}} > 15$ GeV.

- Channel 2b (340 000 SM events):

$$E_{\mathrm{T}}^{\mathrm{miss}} \geq 50 \text{ GeV}, \quad H_{\mathrm{T}} \geq 50 \text{ GeV}, \tag{4}$$

and at least 2 muons/electrons with $p_{\mathrm{T}} > 15$ GeV.

- Channel 3 (8 500 000 SM events):

$$H_{\mathrm{T}} \geq 600 \text{ GeV}, \quad E_{\mathrm{T}}^{\mathrm{miss}} > 100 \text{ GeV}. \tag{5}$$

Each channel contains different BSM processes, which can be found in Tab. 3.

## 2.2 Description of the data format

The generated MC data is stored in the form of ROOT files (including all stable hadrons) and in CSV files including only the information as described above. The CSV files are published on Zenodo (see Tab. 4) and the ROOT files are available upon request.

In the CSV files, each line has variable length and contains 3 event-specifiers, followed by the kinematic features for each object in the event. The line format is:

```
event ID; process ID; event weight; MET; METphi; obj1, E1, pt1, eta1,
            phi1; obj2, E2, pt2, eta2, phi2; ...
```

Table 3: BSM processes in each channel. More information on the masses of the specific processes can be found in the main text.

| BSM process | Channel 1 | Channel 2a | Channel 2b | Channel 3 |
|---|---|---|---|---|
| $Z'$ + monojet | × | × | | × |
| $Z' + W/Z$ | | | | × |
| $Z'$ + single top | × | | | × |
| $Z'$ in lepton-violating $U(1)_{L_\mu - L_\tau}$ | | × | × | |
| $\not{R}$-SUSY stop-stop | × | | × | × |
| $\not{R}$-SUSY squark-squark | × | | | × |
| SUSY gluino-gluino | × | × | × | × |
| SUSY stop-stop | × | | | × |
| SUSY squark-squark | × | | | × |
| SUSY chargino-neutralino | | × | × | |
| SUSY chargino-chargino | | | × | |

The `event ID` is an event specifier. It is an integer to identify the generation of that particular event, included for debugging and reproducibility purposes. The `process ID` is a string referring to the process that generated the event, as mentioned in Tabs. 2 and 3. The event weight $w$ is defined as

$$w = \frac{\sigma}{N_{\text{lines}}} \times \left(10\,\text{fb}^{-1}\right), \tag{6}$$

with $\sigma$ the cross section for a particular process (expressed in fb), and $N_{\text{lines}}$ the number of events in a single CSV file.

Concerning the kinematic features, the `MET` and `METphi` entries are the magnitude $E_{\text{T}}^{\text{miss}}$ and the azimuthal angle $\phi_{E_{\text{T}}^{\text{miss}}}$ of the missing transverse energy vector of the event. The $E_{\text{T}}^{\text{miss}}$ is based on the truth $E_{\text{T}}^{\text{miss}}$, meaning the transverse energy of those objects that genuinely escape detection, such as neutrinos and weakly-interacting new particles. The object identifiers (`obj1`, `obj2`,...) are strings identifying each object in the event, using the identifiers of Tab. 1. Each object identifier is followed by 4 comma-separated values fully specifying the 4-vector of the object: `E1`, `pt1`, `eta1`, `phi1`. The quantities `E1` and `pt1` respectively refer to the full energy $E$ and transverse momentum $p_{\text{T}}$ of `obj1` in units of MeV. The quantities `eta1` and `phi1` refer to the pseudo-rapidity $\eta$ and azimuthal angle $\phi$ of `obj1`.

As an example, an event corresponding to the final state of the $t\bar{t} + 2j$ process with two $b$-jets (with $E = 331.9$ GeV and $E = 55.8$ GeV) and one jet (with $E = 100.4$ GeV) reads:

94;ttbar;1;112288;1.74766;b,331927,147558,-1.44969,-1.76399;j,100406,85589,-
0.568259,-1.17144;b,55808.8,54391.4,-0.198215,1.726

For the hackathon challenge [68], a cocktail of SM backgrounds is provided in four CSV files (one for each channel), with a luminosity of 7.8 fb$^{-1}$ (214 185 events), 309.6 fb$^{-1}$ (20 005 events), 7.8 fb$^{-1}$ (340 268 events) and 8.0 fb$^{-1}$ (8 544 111 events) for channel 1, 2a, 2b and 3,

Table 4: The different datasets and their Zenodo weblinks.

| Dataset | Link | Selection |
|---|---|---|
| Darkmachines generation | [67] | All events in Tab. 2. |
| Unsupervised Hackathon | [68] | Labeled signal and background events. |
| Secret dataset | [69] | Unlabeled dataset. |

respectively. These files have to be split into training data and validation data. The validation background events are supplemented by signal CSV files belonging to the processes summarized in Tab. 3. For channel 1, 2a, 2b and 3 we provide 8 (38 666 events), 6 (5868 events), 9 (89 676 events), and 10 (1 023 320 events) different signal files.

The algorithms are ultimately assessed (see section 2.2.1 for the figures of merit) on their performance on four "secret" datasets [69], one for each channel. These contain a mixture of SM events and non-SM events, where the labels (*i.e.* `process ID`) of the events are hidden. In addition, noise events have been added as an additional blinding mechanism. The events contained in these secret datasets have been generated in a similar way as the training-data events. Note that this is only partially representative of LHC data, as one cannot expect the outcome of the Monte Carlo generation and fast simulation to represent full complexity of the actual physical events that are produced at the LHC, without an estimation of both theoretical and detector-related uncertainties. In any case, and for the scope of this paper, these datasets can be useful to understand the main characteristics and performed of different anomaly detection algorithms under controlled simplified conditions, and we leave a full analysis for data to future studies within the experimental collaborations or using Open Data.

### 2.2.1 Figures of merit

For each event in the validation data and secret dataset an anomaly score is obtained (as detailed in the individual method section in Sec. 4). The receiver operating characteristic (ROC) curve is obtained by scanning over thresholds of the anomaly score to cut on when determining if an event is anomalous or not. The ROC curve is parameterized as the signal efficiency ($\epsilon_S$, the true-positive rate) as a function of the background efficiency ($\epsilon_B$, the false-positive rate). The area under the ROC curve (AUC) is a common metric for classification problems, an AUC of 1.0 is a perfect classifier while a random guess gives an AUC of 0.5. However, the AUC is dominated by the signal efficiency at large background efficiency, whereas many new physics signals need to cut out a much larger fraction of the background. Therefore, we also use as metrics the signal efficiency at three separate working points. The figures of merit (per signal model) used in this study are:

- Area under the Curve (AUC),

- The signal efficiency at a background efficiency of $10^{-2}$, $\left[\epsilon_S(\epsilon_B = 10^{-2})\right]$,

- The signal efficiency at a background efficiency of $10^{-3}$, $\left[\epsilon_S(\epsilon_B = 10^{-3})\right]$,

- The signal efficiency at a background efficiency of $10^{-4}$, $\left[\epsilon_S(\epsilon_B = 10^{-4})\right]$.

In addition, we derive combined performance figures (see section 5) in order to quantify the mean, maximum and minimum performance of the algorithms for the many examined signals.

## 3 Approaches to the problem

Depending on the availability of labelled data, several approaches for the tagging of the signal events are possible. We can summarize at least four of them as follows.

(a) Training the algorithm on computer-generated (simulated) backgrounds. It will then be tested on real data.

(b) Training the algorithm on real data, possibly being a mixture of signal and background. This is necessary when a reliable or accurate model for the background is not available. It will then be tested on another independent sample of real data.

(c) Training the algorithm by two-sample comparison of background data and real data [18, 20]

(d) Training the algorithm on a specific signal and background. This is what is typically done at the LHC. Another possibility would be to train the algorithm on a large number of possible signals with a large variety.

In this challenge we follow the first route, meaning that the anomaly detection algorithm can be trained on a pure SM sample. In this step the algorithm is supposed to learn background specific properties. The trained algorithm is then exposed to a mock dataset where signals of various kinds are injected on top of the SM background. This is done to validate the algorithm and assess its performance to spot outliers.

In all four cases, including ours, the outcome can be reduced to constructing one or more variables which maximize the power to discriminate signal from background (e.g. the probability of being an outlier, see Fig. 9). For all those approaches requiring a background-only dataset as input, one should keep in mind that even the most accurate simulation tools do not provide a perfect description of LHC collisions and the consequent mismodeling may show up as fake new physics signals.

In Sec. 4 we describe all the ML algorithms that have been applied to the challenge. These include Kernel Density Estimation (KDE) [70], Gaussian Mixture Models (GMMs) [71], flow models [72], (variational) autoencoders [73, 74] and Generative Adversarial Networks (GANs) [115]. Although these algorithms are very different from each other, they rely on one or more of the following ingredients: assessing an anomaly score, performing clustering, dimensional reduction and/or density estimation. We dedicate the remaining part of this section to the description of each of these ingredients.

## 3.1 Assessing Anomaly Scores

We present an instructive toy example in Fig. 9. We simulated data from a background expectation distributed exponentially and we combined it with a narrow Gaussian signal anomaly. In order to give an anomaly score to the points we trained the Local Outlier Factor (LOF) [75] on a background-only simulation, and subsequently used it on the dataset containing both inliers and outliers (this would correspond to approach (a) mentioned above). Despite its simplicity, this example shows two interesting characteristics. First, it is clear that feature selection is important, since the variable on the $x$-axis is discriminating, while the variable on the $y$-axis is less discriminating. This is because the exponential distribution of the background has a different variance in the two directions. Second, the example has the characteristics that it is difficult to separate an anomaly from the background with a simple selection on one of the two plotted variables. The purpose of anomaly detection in this context is not to find *all* anomalous points, but to be able to reliably state when a point (or a set of points) is anomalous and worth studying. The LOF gives a score to all points in order to assess how much they differ from the background. On the right-hand side of Fig. 9, we see that most outliers have a high probability of being part of the signal, and not belonging to the background.

Once all points are assigned an anomaly score, one may compare the distribution of such scores to a validation set containing only SM events. Therefore, we use the framework of a two-sample test, aimed at detecting statistically significant differences in the score distributions of inliers and outliers. While this example makes it seem like outliers can only be found in the

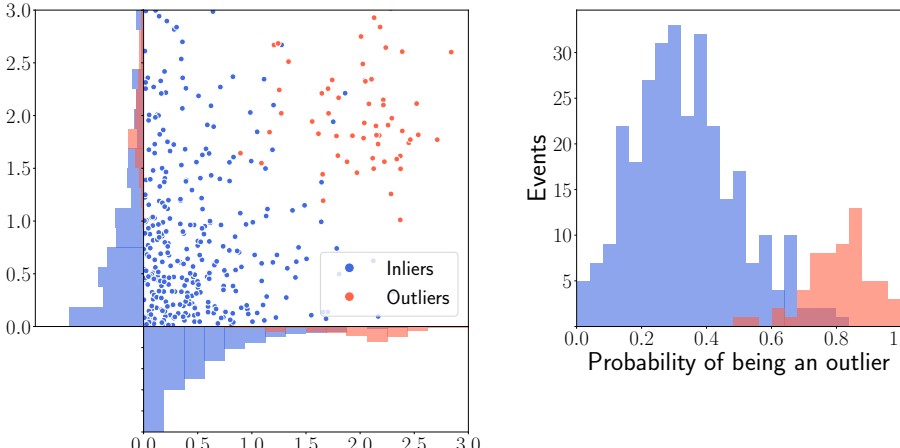

Figure 9: *Left:* A narrow Gaussian anomaly centered around $(2, 2)$ (in red) is added to an exponentially-distributed background (in blue). *Right:* The probability of belonging to the signal events (outliers) is assigned to each point of the dataset and we can perform a counting. In this case, higher probabilities are correctly assigned to the outliers.

tails of distributions, most methods studied in this work could also find anomalous signals in "voids" in the high-dimensional space, up to questions of topology [76].

## 3.2 Clustering

We expect data amenable for analysis to lack in class labels (e.g. it is not known if the data is a signal event); it will then be necessary to extract information in an unsupervised fashion. A solution is to invoke clustering techniques [77, 78], where the goal is to group the data into clusters, each cluster bearing certain unique properties. Specifically, the goal is to partition the data such that the average distance between objects in the same cluster (the average intra-distance) is significantly less than the distance between objects in different clusters (the average inter-distance). Several approaches have been developed to cluster data based on diverse criteria, such as the cluster representation (e.g. flat, hierarchical), the criterion function to identify sensible clusters (e.g. sum-of-squared errors, minimum variance), and the proximity measure that quantifies the degree of similarity between data objects (e.g. Euclidean distance, Manhattan norm, inner product). Our goal is to experiment with a variety of clustering approaches to gain a better understanding of the type of patterns emerging from clustering structures.

In order to analyze clusters to identify novel groupings that may point to new physics, one approach is to use what is known as *cluster validation* [79], where the idea is to assess the value of the output of a clustering algorithm by computing statistics over the clustering structure. Clusters with a high degree of *cohesiveness*, where events within the group are sampled from regions of high probability density, are particularly relevant for analysis. In addition, one could carry out a form of *external cluster validation* [80], where the idea is to compare the output clusters to existing, known classes of particles. While finding clusters resembling existing classes may serve to confirm existing theories, clusters bearing no resemblance to known classes can potentially drive the search for new physics models.

## 3.3   Dimensional Reduction

Data stemming from the LHC arrive in copious amounts, are high-dimensional, and lack class labels; clustering can be useful to find patterns hidden in the data, a task whose importance has been highlighted in the previous section. Unfortunately, high-dimensional data creates a plethora of complications during the data analysis process. Two possible solutions exist: we can either pre-process the data through dimensionality reduction techniques [81], or we can make use of specialized approaches [82].

Dimensionality reduction can be done through feature selection, by determining which features are most relevant, i.e. those that possess a high power to discriminate signal from background. This may come with some information loss, but it is commonly the case at the LHC that only a subset of information is needed to distinguish among different types of data. Another approach is to invoke principal component analysis: the data is transformed while eliminating cross correlations among the new features; the resulting subset can be further analyzed to filter out irrelevant features.

Another promising direction is to use ML to attain a reduced representation of the data by performing non-linear transformations [83, 84]. This approach can have a strong impact in the search for new physics since it implements data transformations that can unveil hidden patterns corresponding to new particle signals.

## 3.4   Density estimation

Events produced at the LHC (either real or simulated) can be thought of as samples drawn from an unknown probability density function (PDF) that characterizes the complex physical processes leading to the generation of the events themselves. The PDF of a new physics signal might be different from the PDF of the SM. However, also the estimated PDFs of the SM, and the one from real experimental data may be different. Spotting and analyzing the differences in these two densities can provide a great deal of information about the underlying process (i.e. the true physical model) that generates the signal events.

Assuming density estimation can be performed accurately, there are several ways to use it for model independent unsupervised analysis. For instance, one can compare the PDFs of real and simulated data to detect differences. They point towards interesting signal regions, which can be used in order to guide further scrutiny. In the context of this challenge, since we determine the anomaly score on a event-by-event basis, we cannot rely on a comparison between the two densities. We can still estimate the PDF of the background dataset and use this in order to determine an anomaly score for new events. However, estimating the PDF reliably starting from the raw data is far from trivial, especially if the number of features is high. This constitutes an active field of research in data science and, depending on the specific task, different approaches may be suitable [70, 85]. One such approach is kernel density estimation, which estimates the PDF by a sum of kernel functions (e.g. multivariate Gaussians) centered around each data point [86]. Furthermore, one could also perform clustering and anomaly detection in a way independent from the approaches mentioned before, see e.g. [87, 88].

One difficulty in applying density estimation on the dataset described in this work is the fact that the events change in dimensionality because the number of objects is not the same in every event. Additionally, there are both continuous data (for example energy and angles) and categorical data (object symbol). To circumvent these issues, one might try to map events to a different parameter space. A potential methodology is described in Ref. [26].

Table 5: Summary of the algorithms.

| Abbreviation | Algorithm | Section | Hyperparameters | # Submitted |
|---|---|---|---|---|
| SimpleAE | Autoencoders | 4.1 | Tab. 6 | 1 |
| VAEs | Variational Autoencoders | 4.2 | Tab. 7 | 140 |
| DeepSetVAE | Deep Set Variational Autoencoders | 4.3 | Tab. 8 | 4 |
| ConvVAE (NoF) | Convolutional Variational Autoencoders | 4.4 | Tab. 9 | 1 |
| Planar | ConvVAE+Planar Flows | 4.5.1 | Tab. 10 | 1 |
| SNF | ConvVAE+Sylvester Normalizing Flows | 4.5.2 | Tab. 11 | 3 |
| IAF | ConvVAE+Inverse Autoregressive Flows | 4.5.3 | Tab. 12 | 1 |
| ConvF | ConvVAE+Convolutional Normalizing Flows | 4.5.4 | Tab. 13 | 1 |
| CNN | Convolutional ($\beta$)VAE | 4.6 | | 2 |
| KDE | Kernel Density Estimation | 4.7 | Tab. 14 | 36 |
| Flow | Spline autoregressive flow | 4.8 | Tab. 15 | 2 |
| Deep SVDD | Deep SVDD | 4.9 | Tab. 16 & 17 | 80 |
| Combined (Deep SVDD & Flow) | Spline autoregressive flow with Deep SVDD | 4.10 | | 8 |
| DAGMM | Deep Autoencoding Gaussian Mixture Model | 4.11 | Tab. 19 | 384 |
| ALAD | Adversarial Anomaly Detection | 4.12 | Tab. 21 | 96 |
| Latent | Anomaly Detection in the Latent Space | 4.13 | Tab. 22 | 288 |

# 4 Methods

In this section we described the employed methods. For every method, we also indicated the authors of that section with a footnote. The methods are summarized in Tab. 5, where the number of submitted models (i.e. with different sets of hyperparameters) within that category is also quoted.

## 4.1 Autoencoders[9]

Autoencoders (AEs) [89] are a class of deep neural networks characterised by a central hidden layer of lower dimension than the input layer, and a target space coinciding with the input space. AEs can therefore be trained to reconstruct the various features of the input data, while their bottleneck structure prevents them from simply learning the identity map. The dimension of the central layer is of particular importance, as it determines the amount of compression of the features between the input space and this latent space. The AE architecture can thus be deconstructed into an encoder part, which compresses the input data into the latent space, and a decoder part, which uses information from the lower dimensional latent space to extrapolate to the full input space. A promising use of AEs in HEP has been outlined in Refs. [90, 91], namely the online compression of jets during data-taking at the LHC and their

---

[9]Baptiste Ravina, Marija Vaškevičiūte, Erik Wulff, Honey Gupta, Erik Wallin, Jessica Lastow, Antonio Boveia, Lukas Heinrich, Caterina Doglioni

Table 6: Hyperparameters and input features for the benchmark model, as well as the model optimised specifically for Channel 2a after a grid search.

| Parameter | Benchmark model | Optimised model |
|---|---|---|
| activation | tanh | ReLU |
| dropout | None | 0.05 |
| kernel init. | normal | uniform |
| latent dim. | 10 | 15 |
| # of layers | 3 | 1 |
| optimiser | Adam | SGD |
| input features | leading 8 objects | leading 4 objects |
| | + missing $E_T$ | + missing $E_T$ |

high-fidelity offline decompression, offering competitive processing rates and a reduced need for data storage. Furthermore, an AE trained extensively on SM data and reaching arbitrarily low reconstruction errors could be used as an anomaly detection algorithm. Presented with new data, the AE could tag anomalous events from their comparatively higher reconstruction error.

Here we consider an AE architecture inspired by the results of Refs. [90, 91], shown to perform well on the identity reconstruction of an independent dataset of dijet events, currently being studied for data compression. This benchmark model is left unoptimised with respect to the various data channels of the challenge at hand. Only in Channel 2a, where the available training statistics are insufficient to achieve a similar performance as in the other channels, is a simple grid search performed over a small range of hyperparameters. The benchmark model consists of an encoder with three hidden dense layers with 200, 200 and 20 nodes respectively, leading to a latent layer with 10 nodes; the structure of the decoder is completely symmetric with respect to the encoder. No dropout is applied anywhere and all weights are initialised from a normal kernel. A tanh activation function is applied after each inner layer, while the output layer receives linear activation. One instance of the model is trained per channel of the dataset, reserving 10% of the training data for validation and considering 125 events per batch. The Adam optimiser [92] is used with an MSE loss function. An early stopping strategy is adopted, ending the training when no improvement in the validation loss is observed after 20 consecutive epochs.

Data are pre-processed as follows: only the 4-vectors $(p_T, \eta, \phi, E)$ of the leading 8 objects in $p_T$ are kept, together with the magnitude and azimuthal component of the missing transverse energy. Object labels are not considered. Where there are fewer than 8 objects in an event, zero-padding is applied. All features are then standardised over the entire training dataset; the same means and standard deviations are then used to standardise subsequent datasets (validation and test signals) in a consistent manner. Tab. 6 summarises the structure of this benchmark model, and highlights differences found in the optimisation of the specific model targeting Channel 2a.

## 4.2 Variational autoencoders[10]

Variational Autoencoders (VAEs) [74] are a class of autoencoder architecture (sec. 4.1) where the output is equal to the input, and the bottleneck layer is generated by letting the encoder output two numbers per latent space dimension, which represent a mean and standard deviation for a normal distribution. A sample is drawn from this set of distributions and the sample

---

[10]Luc Hendriks, Roberto Ruiz

is run through the decoder to reconstruct the original input. A term is added to the loss function proportional to the the Kullback-Leibler (KL) divergence, which tries to make the latent space normally distributed.

It is generally thought that the reconstruction loss of a VAE is a good anomaly score variable. The VAE has to compress the original input data into a lower dimensional representation, and so it has to efficiently store the relevant data in the latent space in order to be able to reconstruct the input data well at the output stage. Signal events look different from background events, and as the VAE has not been optimised for this, they will be reconstructed more badly than the background events. To balance out the relative importance of the reconstruction loss and the KL-divergence term, a term $\beta$ is sometimes introduced as a hyperparameter to control this relative importance. The loss function then becomes

$$L_{\text{VAE}} = (1 - \beta)\text{MSE} + \beta\text{KL}. \tag{7}$$

Not only the reconstruction loss can be a good outlier variable. Because the KL-divergence favors inputs that are encoded near the center of the latent space, the radius from the center is another anomaly score definition that can have predictive results. We refer to the VAE using this form of the loss function as the $\beta$-VAE.

Table 7: All hyperparameters used to train the various $\beta$-VAE models.

| Parameter | Values |
|---|---|
| $\beta$ | $[10^{-6}, 10^{-3}, 0.1, 0.5, 0.8, 0.999, 1]$ |
| z | $[5,8,13,21,34]$ |
| Anomaly score | [Reconstruction loss, Radius] |
| Dataset | [Dynamic, Static] |

We try different hyperparameter combinations and anomaly score definitions and compare their performance. We performed a grid search with all of the combinations from Tab. 7. The dataset is preprocessed in two ways: a *dynamic* and a *static* method. The dynamic dataset orders the objects in an event by $p_{\text{T}}$ and contains both the object type as class variables and the object properties and $E_{\text{T}}^{\text{miss}}$ and $\phi_{E_{\text{T}}^{\text{miss}}}$ as regression variables. The loss function is updated to contain parts for the classification and regression parts for the reconstruction. This setup is equivalent to the method described in 4.13. In the static dataset the object type is implicit in the object ordering. We take for every object in the dataset the maximum number of those in the entire dataset and add a label if the particular object is in a particular event. In this way, an event will look like

$$\begin{aligned}
\vec{x} \quad = \Big( & E_{\text{T}}^{\text{miss}}, \phi_{E_{\text{T}}^{\text{miss}}}, \quad x_{j,1}, x_{j,1,p_{\text{T}}}, x_{j,1,\eta}, x_{j,1,\phi}, \dots, \\
& x_{j,n_j}, x_{j,n_j,p_{\text{T}}}, x_{j,n_j,\eta}, x_{j,n_j,\phi} \dots, \\
& \text{for all object types} \Big).
\end{aligned} \tag{8}$$

The advantage is that there is no classification part necessary in the $\beta$-VAE, as the object type is defined explicitly by its position in the input vector.

Note that when $\beta = 1$, the only relevant part in the loss function is the KL-divergence term, effectively rendering the decoder useless, as there is nothing in the loss function that pushes the weights in the decoder to particular values.

### 4.3 Deep set variational autoencoder[11]

The main idea behind this method is that the outgoing particles in collider experiments can be thought of as a collection of four-vectors. There is no intrinsic ordering, although we often sort them by the magnitude of the transverse momentum. Therefore, using a network architecture which respects the permutation symmetry is more natural and could lead to improved results. In Ref. [93], it was shown that "deep sets" with permutation-invariant functions of variable-length inputs can be parameterized in a fully general way. This idea was introduced to the High Energy Physics community in Ref. [94], where the operations were generalized to include infrared and collinear safety. While their methods outperformed other state-of-the art classifiers, they found a slight improvement if the operations were not restricted to be IRC safe.

For an unsupervised learning task, we modify the deep sets paradigm to include an auto encoding structure, following the example of Ref. [95]. As in Ref. [94], we map each particle to the latent space using a common function $\Phi$. The functional form of $\Phi$ is a four layer network which has inputs of $(\log_{10} E, \log_{10} p_{\mathrm{T}}, \eta, \phi, \mathrm{PID})$.

One possible way of combining the per-particle latent spaces into an event-level latent representation is to sum the individual components [95]. However, it was shown that sorting each feature (for instance all of the first latent dimension) along all of the particles, followed by learned mapping from the sorted features to the latent space improves performance. This layer/operation is known as FSPool [96]. After the FSPooling layer, we reparameterize the system using a variational autoencoder with a Gaussian prior.

A decoding network is then used to transform the latent data back to a set of four-vectors and particle IDs. We do so using a dense neural network. We use two layers with 256 nodes with ReLU activations. From here, the network splits to 80 nodes, representing the 20 four-vectors, and a series of 180 nodes representing the probabilities that each of the 20 particles belongs to one of the eight particle IDs or be masked out (as in there should not be another particle).

Once the final set is obtained, we can compute the loss compared to the initial set. This is made more challenging because of the permutation invariance–the first input particle does not correspond to the first output particle. We therefore use a modified version of the Chamfer loss, which is given by

$$L_{\mathrm{C}} = \sum_{x \in S_{\mathrm{input}}} \min_{y \in S_{\mathrm{output}}} \left| \vec{x} - \vec{y} \right| + \sum_{y \in S_{\mathrm{output}}} \min_{x \in S_{\mathrm{input}}} \left| \vec{x} - \vec{y} \right|. \tag{9}$$

In addition to this, we include the $-\log \mathrm{SoftMax}$ for the PID prediction of $y$ for the true class of $x$ for the pair which minimizes the distance. It is unclear how to weight this classification loss compared to the distance loss, so we experiment with different values. In addition, we test the ratio of the loss of the KL divergence of the latent space and the Chamfer loss similar to Eq. (7), substituting MSE by $L_C$. Thus the total loss is given by

$$L_{\mathrm{DeepSet}} = (1 - \beta) \, L_{\mathrm{C}} + \beta \, \mathrm{KL}. \tag{10}$$

The parameters used for this study are summarized in Tab. 8. Note that the values of $\beta = 0$ or $\beta = 1$ performed best, so only these were submitted to the challenge for further analysis. More details on this method can be found in Ref. [97].

### 4.4 Convolutional variational autoencoder[12]

In this method we use a one class trained VAE with the reconstruction loss serving as the anomaly score. The VAE is only trained to reconstruct background events and as a result,

---

[11]Author: Bryan Ostdiek

[12]Pratik Jawahar, Maurizio Pierini, Kinga Anna Wozniak, Mary Touranakou, Javier Mauricio Duarte

Table 8: All hyperparameters used to train DeepSet $\beta$-VAE models.

| Parameter | Values |
|---|---|
| $\beta$ | $[0, 10^{-6}, 10^{-3}, 0.1, 0.5, 0.8, 0.999, 1]$ |
| Latent space dimension | 8 |
| Encoder Width | 256 |
| Decoder Width | 256 |
| Weighting of particle ID prediction | $[1, 10]$ |
| Anomaly score | Total Loss [Eq. (10)] |

Table 9: ConvVAE hyper-parameters.

| Parameter | Values |
|---|---|
| learning rate | $(10^{-3}$ with decay$)$ |
| batch size | 32 |
| latent space dimension | 10 |
| kernel size | $[(3,4),(5,1),(7,1)]$ |
| stride | 1 |
| anomaly score | Chamfer Loss [Eq. 9] |

signal events yield a higher reconstruction loss. Applying a set threshold, we classify events with a reconstruction loss higher than the threshold as signal events and the rest as background events. The architecture used here is a ConvVAE [98], in which the encoder and decoder are composed of Convolutional Neural Networks. We pre-process the input into an image-like 2D matrix, such that the CNN identifies information such as number of objects of each type per event as spatial features. The "image" is a $4 \times n$ matrix where the 4 rows are $[E, p_T, \eta, \phi]$ for each of the $n$ objects. The loss function used is composed of the KL Divergence term and the Chamfer Loss term defined in Eq.(9) by choosing $\beta = 1$. This method relies on good reconstruction of background events since the model is only trained on this class, and expects poor reconstruction of signal events to allow simple threshold-on-loss strategies for anomaly detection. The hyperparamters used are shown in Tab. 9.

## 4.5 ConvVAE with normalizing flows[13]

With the ConvVAE of section 4.4 serving as the baseline for this method, we optimize performance in anomaly detection using normalizing flows to learn a better suited posterior approximation [99], in place of the multivariate normal approximation made in the baseline ConvVAE model. The same input format as Sec 4.4 is used.

A normalizing flow can be generalized as any invertible transformation that can be applied to a given distribution to generate a desired target distribution. In order to be compatible with variational inference, it is desirable for the transformations to have an efficient mechanism for computing the determinant of the Jacobian, while being invertible [99]. We utilize 4 major families of flow models described below, to learn better approximate posteriors as part of a single, sequential training process.

---

[13]Pratik Jawahar, Maurizio Pierini, Javier Mauricio Duarte

### 4.5.1 Planar flows

Planar flows were introduced in Ref. [99] as invertible transformations whose Jacobian determinant can be computed rather efficiently, making them suitable candidates to be used in variational inference. Planar flow transformations are defined as,

$$\mathbf{z}' = \mathbf{z} + \mathrm{u}h(\mathrm{w}^T\mathbf{z} + b). \tag{11}$$

Here, $\mathrm{u}, \mathrm{w} \in \mathbb{R}^D$, $b \in \mathbb{R}$ and $h(\mathbf{z})$ is a suitable smooth activation function. The additional hyper parameters used are shown in Tab. 10.

Table 10: Planar model hyper-parameters.

| Parameter | Values |
| --- | --- |
| flow layers | 6 |
| dense layers per flow layer | 3 |
| neurons per dense layer | 90 |

### 4.5.2 Sylvester normalizing flows

Sylvester normalizing flows (SNFs) [100] build on the planar flow formulation and extend it to be analogous to a multi layer perceptron with one hidden layer of $M$ units and a residual connection as,

$$\mathbf{z}' = \mathbf{z} + \mathbf{A}h(\mathbf{B}\mathbf{z} + b). \tag{12}$$

Here, $\mathrm{A} \in \mathbb{R}^{D \times M}$, $\mathrm{B} \in \mathbb{R}^{M \times D}$, $b \in \mathbb{R}^M$ and $M \leq D$. Computing the Jacobian determinant for such a formulation is made more efficient by utilizing the Sylvester determinant identity [100]. Depending on the way $A$ and $B$ are parametrized, we get different types of SNFs. In this paper we use orthogonal, Householder, and triangular SNFs. The model parameters used are shown in Tab. 11.

### 4.5.3 Inverse autoregressive flows

Autoregressive transformations are invertible learnable functions, hence a suitable choice to define a normalizing flow. However, computing the transformation requires multiple sequential steps [100]. The inverse transformation however, leads to certain simplifications allowing more efficient parallel computing, thereby making it a more desirable transformation for our case. Thereby, we use the inverse autoregressive flows (IAF) [72] formulated as,

$$z_i^t = \mu_i^t(z_{1:i-1}^{t-1}) + \sigma_i^t(z_{1:i-1}^{t-1}) \cdot z_i^{t-1}, \quad i = 1, 2, \ldots, D, \tag{13}$$

where $t$ is the number of IAF transformations applied and $D$ is the number of latent dimensions. Such a formulation allows stacking of multiple transformations to achieve more flexibility in producing target distributions. The model parameters used are shown in Tab. 12.

### 4.5.4 Convolutional normalizing flows

Convolutional normalizing flows (ConvF) [101] extrapolate the idea of planar flows with a single hidden unit [72] to multiple hidden units and replace the fully connected network operation with a 1-D convolution to achieve bijectivity giving,

$$\mathbf{z}' = \mathbf{z} + \mathbf{u} \odot h(\mathrm{conv}(\mathbf{z}, \mathbf{w})). \tag{14}$$

Table 11: SNF model hyper-parameters.

| Parameter | Values |
|---|---|
| flow layers | 4 |
| dense layers per flow layer | 5 |
| no. of orthogonal vectors | 8 |
| no. of Householder transformations | 8 |

Table 12: IAF model hyper-parameters.

| Parameter | Values |
|---|---|
| MADE layers | 4 |
| MADE neurons per layer | 330 |

Table 13: ConvF model hyper-parameters.

| Parameter | Values |
|---|---|
| flow layers | 4 |
| flow kernel size | $(7, 1)$ |
| dilation | True |

Here, $w \in R^k$ is the parameter of the 1-D convolution filter with $k$ being the kernel width; $h$ is a monotonic non-linear activation function and $\odot$ denotes point-wise multiplication [101]. The model parameters used are shown in Tab. 13.

## 4.6 Convolutional $\beta$-VAE[14]

As for the VAEs in sections 4.2 and 4.3, we investigate the impact of a $\beta$ term in the loss definition of the ConvVAE, defined according to Eq.(7). On investigation of the CNN-VAE approach (sec. 4.4), we found some issues with the KL divergence pulling reconstructions towards a Gaussian distribution, rather than the true distribution of the input data. In some cases this can be solved by minimizing the effect of the KL divergence by adding a tweak-able $\beta$ term. The $\beta$-VAE has an emphasis on discovering disentangled latent factors. If each variable in the latent space is only sensitive to one generative factor, this representation is considered disentangled. This can lead to greater interpretability of the model and generalization to a wider breadth of tasks.

The $\beta$ hyperparameter acts as a Lagrange multiplier, and looks to aid in optimization towards a local minimum [102]. The values range between 0 and 1 and we adjust both elements of the loss function by a scale that allows the contribution to match each other, keeping the co-dependence of the loss functions.

The values of $\beta$ are different depending on the channel and trial and error was used to find which value performed best for each model. Channel 1: $\beta = 0.4$, Channel 2a: $\beta = 0.9$, Channel 2b: $\beta = 0.3$, Channel 3: $\beta = 0.1$.

---

[14]Author: Joe Davies

Table 14: KDE model hyper-parameters.

| Parameter | Values |
|---|---|
| dimensionality reduction method | [PCA, VAE] |
| $\beta$ | $[10^{-5}, 10^{-3}, 0.1]$ |
| latent space dimension $D$ | $[2, 3, 4, 5, 6, 7, 8, 9, 10]$ |

## 4.7  Kernel density estimation[15]

A simple yet powerful approach to the task of finding anomalous events is given by density estimation. Starting from the background-only sample, the PDF reconstructed from these points can be estimated as $\hat{p}_b$ using kernel density estimation (KDE). Then the events that appear as rare will be considered anomalous: for a given event $x$, an anomaly score can be defined as

$$S(x) = -\log \hat{p}_b(x). \tag{15}$$

However, estimating the PDF of the background is not a trivial task. The first issue arises from the curse of dimensionality. Our input dataset contains the missing energy information and an ordered sequence of 4-momenta with zero-padding. In particular the objects whose 4-momenta we consider are the 10 leading jets, 4 bottoms, 3 of each lepton type and 2 photons, which means that the input dataset has around 100 features. In order to overcome this issue, we perform dimensionality reduction in different ways. We either use principal component analysis (PCA) [103] or a $\beta$-VAE whose complexity is comparable to PCA. This simple $\beta$-VAE consists of an encoder and decoder with a hidden layer of 32 nodes and a bottleneck size $D$. The loss is given by equation 7.

The methods used for density estimation will differ in the way we perform the dimensionality reduction (PCA or $\beta$-VAE). In the case of the $\beta$-VAE, our analysis shows that small values of $\beta$ lead in general to better results with the density estimation approach. Our methods are also characterized by the final number of features $D$. This is summarized in Tab. 14.

KDE requires longer computation times for an increasing number of samples. This means that, depending on the channel, we will use different procedures. For channels 1 and 2a, we perform a 5-fold cross-validation in order to assess the optimal bandwidth, based on the data-based maximum likelihood. Both the cross-validation and the KDE are performed with the scikit-learn libraries [104].

However, channels 2b and 3 have a larger sample size and this makes the previous procedure unfeasible. In this case, we adopt as the optimal bandwidth choice the one from Silverman's rule of thumb [105]. By looking at the results for channels 1 and 2a we know that the optimal bandwidth found with this rule is close to the one found from cross-validation. Then, the density estimation is performed using fast Fourier transforms on a grid, which is already implemented in KDEpy [106] and makes the computations considerably faster. Finally, in order to assess the anomaly score of an event, we use a nearest neighbor interpolation with weights inversely proportional to the distances. These weights are used with the aim of lifting residual degeneracies in events sharing the same nearest neighbors.

## 4.8  Spline autoregressive flows[16]

While in Sec. 4.5 normalizing flows are used as a posterior for a ConvVAE, they may also be applied in isolation. In [107] an autoregressive flow model was used to infer the likelihood

---

[15]Andrea De Simone, Alessandro Morandini
[16]Luc Hendriks, Rob Verheyen

of HEP events from weighted training data, with the goal of being able to sample new events from the model. However, using the tractable likelihood of normalizing flows, this model may also be used as an anomaly detector. While the model is similar to the normalizing flows mentioned earlier, the most significant difference is that, instead of the relatively simple transforms used previously, this model uses rational quadratic splines (RQS). These are highly expressive functions with well-defined domains, which is particularly useful for HEP events as they fill a bounded phase space. The RQS transforms are parameterized by MADE networks [108], which are autoregressive neural networks that ensure efficient tractability of the flow likelihood.

The anomaly score of an event $x$ is defined as

$$S(x) = \frac{\log p(x) - \log p_{\min}}{\log p_{\max} - \log p_{\min}}, \tag{16}$$

where $p(x)$ is the flow likelihood, and $p_{\min}$ and $p_{\max}$ are respectively the minimum and maximum likelihoods to appear among the evaluated event samples.

The flow model parameters are defined in Tab. 15. The dataset is parsed in a few different configurations:

- **Efficient**: For every event, only the $E_T^{\text{miss}}$, $\phi_{E_T^{\text{miss}}}$, number of every object type and the $E$, $p_T$, $\eta$ and $\phi$ of the top 7 jets, or $b$-jets and top 4 leptons,

- **Efficient no E**: For every event, only the $E_T^{\text{miss}}$, $\phi_{E_T^{\text{miss}}}$, number of every object type and the $p_T$, $\eta$ and $\phi$ of the top 7 jets, or $b$-jets and top 4 leptons,

- **Only Aggregates**: For every event, only the $E_T^{\text{miss}}$, $\phi_{E_T^{\text{miss}}}$ and number of every object type.

In the result tables, these models are indicated by *Flow-algorithmName_Likelihood*.

Table 15: Parameter combinations used for training the spline autoregressive flow model.

| Hyperparapeter | Value |
|---|---|
| Initial learning rate | 0.001 |
| Batch size | 512 |
| Optimizer | Adam [109] |
| Loss function | Log-likelihood |
| RQS knots | 35 |
| Flow layers | 11 |
| MADE layers | 7 |
| MADE neurons per layer | 200 |
| Epochs (channel 1, 2a, 2b) | 100 |
| Epochs (channel 3) | 10 |

## 4.9 Deep SVDD models[17]

Deep Support Vector Data Description (SVDD) models [110] are neural networks that go from an input to a vector of constant numbers. The loss function of the neural network is simply

---

[17]Luc Hendriks, Sascha Caron

Table 16: Parameter combinations used for training the Deep SVDD models.

| Hyperparameter | Value |
|---|---|
| Initial learning rate | 0.001 |
| Batch size | 10000 |
| Optimizer | Adam [109] |
| Loss function | Mean squared error |
| Dense layers | 3 |
| Neurons per layer | [512, 256, 128] |

Table 17: Network parameters of the Deep SVDD networks.

| Parameter | Values |
|---|---|
| Output values | [0, 1, 2, 3, 4, 10, 25] |
| Output value dimensionality | [5, 8, 13, 21, 34, 55, 89, 144, 233] |

the mean squared error of the predicted values versus the expected values, which is the same value for all inputs. The expected output value and the dimensionality of the vector are hyperparameters. The loss function is given by equation 17, where $d$ is the length of the output vector and $C_d$ is the output value:

$$\mathcal{L} = \frac{1}{d}|\hat{y} - C_d|^2.$$ 

(17)

In addition to this standard deep SVDD approach we also trained a set of networks using the KL divergence loss of the network output and a standard normal distribution. In the result tables the algorithms are labelled as follows: *DeepSVDD_CC_dd*, where $C$ and $d$ represent the target value and vector length respectively and the loss function is either MSE or KL. Additionally, there is a run where the MSE is taken over all values C, these are labelled with *DeepSVDD_Reduced_dd* The hyperparameter combinations are shown in Tab. 16 and the chosen values for $C$ and $d$ in Tab. 17.

## 4.10 Spline autoregressive flow combined with deep SVDD models[18]

The Spline autoregressive flow model and Deep SVDD models are also combined to obtain a single score which combines the likelihood approach of the "Flow efficient" model (sec. 4.8) and the combined anomaly score of the Deep SVDD models (Sec. 4.9). The ensemble of Deep SVDD models are first combined using the methods described in Sec. 4.13. Then, the flow model score and combined Deep SVDD model score are combined into a single score using the same method. This method is described in detail in [111]. The results are labelled as *Combined-combination-DeepSVDD-Flow* in the results.

Additionally we also combined the results of the VAE with $\beta = 1$ and $z = 21$ (which was the best performing VAE) with the flow model. This result is labelled as *Combined-combination-VAE_beta1_z21-Flow*. Interestingly, the $\beta = 1$ case behaves similarly to a Deep SVDD model, except that the loss function is not the mean squared error of the network output and a constant output, but the KL divergence of the network output and a standard normal distribution. The $\beta = 1$ VAE is therefore also a "fixed target" neural network.

---

[18]Luc Hendriks, Rob Verheyen, Sascha Caron

## 4.11  Deep Autoencoding Gaussian Mixture Model[19]

The Deep Autoencoding Gaussian Mixture model (DAGMM) [112] combines dimensional reduction performed by a deep autoencoder and density estimation on the learned low-dimensional space.

In the literature this is typically done in a two-step approach due to the difficulty of doing a joint optimization. DAGMM addresses this through a sub-network called an estimation network which basically learns a density in the low-dimensional space generated by the compression network. Thus the DAGMM model consists of a compression network and an estimation network.

*Compression network*. The compression network reduces the dimensionality of the input vector $\mathbf{x}$ through an encoder network to a latent representation $\mathbf{z_c} = E(\mathbf{x}, \theta_e)$, and also reconstructs it to a vector $\mathbf{x}'$ through a decoder network $\mathbf{x}' = D(\mathbf{z}_c, \theta_d)$. Then with the $\mathbf{x}$ and $\mathbf{x}'$ vectors, error features are computed $\mathbf{z}_r = f(\mathbf{x}, \mathbf{x}')$, where $f$ represents multiple distance metrics such as the Euclidean distance, cosine similarity, etc... Here $\theta_e$ and $\theta_d$ are the parameters of the encoder and decoder networks respectively.

*Estimation network*. The goal of the estimation network is to make density estimation with a Gaussian Mixture Model (GMM). It takes as input $\mathbf{z} = [\mathbf{z}_c, \mathbf{z}_r]$ and predicts the $K$ components of the GMM. It is done with a network which outputs $\mathbf{p} = N(\mathbf{z}, \theta_m)$ where $\theta_m$ parametrize $N$. Finally the GMM soft-mixture components vector $\hat{\gamma} = \text{softmax}(\mathbf{p})$.

For a batch of N components the GMM parameters are estimated as follows [112]

$$\hat{\phi}_k = \sum_{i=1}^{N} \frac{\hat{\gamma}_{ik}}{N}, \; \hat{\mu}_k = \frac{\sum_{i=1}^{N} \hat{\gamma}_{ik} \mathbf{z}_i}{\sum_{i=1}^{N} \hat{\gamma}_{ik}}, \; \hat{\Sigma}_k = \frac{\sum_{i=1}^{N} \hat{\gamma}_{ik}(\mathbf{z}_i - \hat{\mu}_k)(\mathbf{z}_i - \hat{\mu}_k)^T}{\sum_{i=1}^{N} \hat{\gamma}_{ik}}, \tag{18}$$

Table 18: DAGMM model architecture. Here d is the number of dimensions of the latent space whereas $d_1$ and $d_2$ are the number of nodes corresponding to the estimation network layers.

| Operation | Units | Activation |
|---|---|---|
| E(x) | | |
| Number of hidden layers | 3 | |
| Dense | 128 | tanh |
| Dense | 256 | tanh |
| Dense | 512 | tanh |
| Output | d | linear |
| D(z) | | |
| Number of hidden layers | 3 | |
| Dense | 512 | tanh |
| Dense | 256 | tanh |
| Dense | 128 | tanh |
| $E_{xz}(x,z)$ | | |
| Number of hidden layers | 1 | |
| Dense | $d_1$ | tanh |
| Output | $d_2$ | softmax |

---

[19]Roberto Ruiz

Table 19: DAGMM model hyper-parameters.

| Parameter | Values |
|---|---|
| learning rate | $[10^{-4}, 10^{-5}]$ |
| number of epochs | $[50, 100]$ |
| batch size | $[100, 500, 1000]$ |
| latent space dimensions d | $[10, 20, 30]$ |
| number of nodes $d_1$ | $[10, 15]$ |
| number of nodes $d_2$ | $[4, 8]$ |
| $\lambda_1$ | $[10^{-2}, 10^{-3}]$ |
| $\lambda_2$ | $[10^{-2}, 10^{-3}]$ |

where $\hat{\phi}$, $\hat{\mu}$ and $\hat{\Sigma}_k$ are the mixture probability, mean and covariance for component $k$ of the GMM, respectively. Finally the sample energy [112] is defined as

$$E(\mathbf{z}) = -log\left(\sum_{k=1}^{K} \frac{exp\left(-\frac{1}{2}(\mathbf{z} - \hat{\mu}_k)^T \hat{\Sigma}_k^{-1}(\mathbf{z} - \hat{\mu}_k)\right)}{\sqrt{det(2\pi\hat{\Sigma}_k)}}\right). \tag{19}$$

The sample energy can be interpreted as the negative log probability associated with $\mathbf{z}$. Thus when minimizing the energy, we assign higher probability to normal data and vice versa for anomalies. We add this value to our loss function and use it as the anomaly score.

Finally the loss function is defined as

$$L = ||\mathbf{x} - \mathbf{x}'||_2 + \lambda_1 E(\mathbf{z}) + \lambda_2 P(\hat{\Sigma}), \tag{20}$$

where the first factor is the well-known $L_2$-norm, $P$ is a function to regularize the singularities caused by zeroes in the covariance matrices (see Ref. [112] for details) and $\lambda_1$ and $\lambda_2$ are hyper-parameters which we optimize.

We consider two features as the outputs of the compression network: the Euclidean distance and the cosine similarity as in Ref. [112]. These plus the latent representation $\mathbf{z}_c$ of the data $\mathbf{x}$ constitute the vector $\mathbf{z}$ which is fed into the estimation network. Therefore the vector $\mathbf{z}$ has dimensions $d + 2$ where $d$ is the latent space dimensionality.

The model architecture is summarized in Tab. 18 to which we add a dropout rate of 0.5 to the estimation network. Regarding the optimization of the model, we have adopted the Adam optimizer and have varied the hyper-parameters as shown in Tab. 19. The data encoding follows the *static* prescription described in 4.2.

## 4.12 Adversarial Anomaly Detection[20]

The adversarial anomaly detection (ALAD) algorithm [113, 114] is a hybrid method which combines generative adversarial networks [115] with autoencoders [74], designed for anomaly detection. Generative adversarial networks (GANs) are composed of two neural networks which compete against each other during training. One network is the generator $G : \mathcal{Z} \to \mathcal{X}$ and the other one is the discriminator $D : \mathcal{X} \to [0, 1]$. While the generator network $G$ learns to generate new samples from a latent space $\mathcal{Z}$, the discriminator one has the role of distinguishing real samples from generated ones. The ALAD GAN is based on BiGANs which inherit implicit regularization, mode coverage, and robustness against mode collapse [116].

---

[20]Roberto Ruiz

Table 20: ALAD model architecture. Here d is the number of dimensions of the latent space.

| Operation | Units | Activation |
|---|---|---|
| E(x) | | |
| Number of hidden layers | 4 | |
| Dense | 64 | leaky ReLU |
| Dense | 128 | leaky ReLU |
| Dense | 256 | leaky ReLU |
| Dense | 512 | leaky ReLU |
| Output | d | linear |
| G(z) | | |
| Number of hidden layers | 4 | |
| Dense | 64 | ReLU |
| Dense | 128 | ReLU |
| Dense | 256 | ReLU |
| Dense | 512 | ReLU |
| $D_{xz}(x,z)$ | | |
| Number of hidden layers | 3 | |
| Dense | 128 | leaky ReLU |
| Dense | 128 | leaky ReLU |
| Dense | 128 | leaky ReLU |
| Output | 1 | sigmoid |
| $D_{xx}(x,\hat{x})$ | | |
| Number of hidden layers | 1 | |
| Dense | 128 | leaky ReLU |
| Output | 1 | sigmoid |
| $D_{zz}(z,\hat{z})$ | | |
| Number of hidden layers | 1 | |
| Dense | 128 | leaky ReLU |
| Output | 1 | sigmoid |

The use of GANs as anomaly detectors has been studied in HEP literature (see e.g. [114]). The particularity of the ALAD method is that it adds an encoder $E : \mathcal{X} \to \mathcal{Z}$ to the GAN [116, 117], besides a discriminator $D_{xz}$ which takes $x$ and $z$ as inputs and is trained simultaneously with the generator. It allows to derive an anomaly-score $A(x)$ by comparing real samples with reconstructed ones by the generator using some metric $(A(x) = f(x, G(E(x)))$. Finally, two additional discriminators $D_{xx}$ and $D_{zz}$ are incorporated to help with the training convergence. With these additions the ALAD objective function becomes (see Ref. [113] for details)

$$\min_{G,E} \max_{D_{xz},D_{xx},D_{zz}} V(D_{xz}, E, G) + V(D_{xx}, E, G) + V(D_{zz}, E, G), \tag{21}$$

where

$$V(D_{xz}, E, G) = \mathbb{E}_{x \sim p_{\mathcal{X}}}[log\, D_{xz}(x, E(x))] + \mathbb{E}_{x \sim p_{\mathcal{Z}}}[log\,(1 - D_{xz}(G(z), z)]\,, \tag{22}$$

$$V(D_{xx}, E, G) = \mathbb{E}_{x \sim p_{\mathcal{X}}}[log\, D_{xx}(x, x)] + \mathbb{E}_{x \sim p_{\mathcal{X}}}[log\,(1 - D_{xx}(x, G(E(x))))] \tag{23}$$

Table 21: ALAD model hyper-parameters.

| Parameter | Values |
|---|---|
| learning rate | $10^{-5}$ |
| number of epochs | 2000 |
| batch size | $[100, 500, 1000, 5000, 10000, 20000]$ |
| latent space dimensions | $[10, 20, 30, 40]$ |

and

$$V(D_{zz}, E, G) = \mathbb{E}_{x \sim p_{\mathcal{Z}}}[log \, D_{zz}(z, z)] + \mathbb{E}_{x \sim p_{\mathcal{Z}}}[log \, (1 - D_{zz}(z, E(G(z))))] . \quad (24)$$

For the anomaly scores we consider the ones used in Ref. [114]

- The $L_1$ distance: $A_{L_1}(x) = ||x - G(E(x)||_1$

- The $L_2$ distance: $A_{L_2}(x) = ||x - G(E(x)||_2$

- A Logits-score: $A_L(x) = log(D_{xx}(x, G(E(x))))$

- A Features-score: $A_F(x) = ||f_{xx}(x, x) - f_{xx}(x, G(E(x))))||_1$

As in the DAGMM case we have employed the Adam optimizer. The ALAD model architecture is shown in Tab. 20. We have further added dropout and batch normalization following Ref. [114]. Finally, a summary of the model hyper-parameters adopted in this work is displayed in Tab. 21.

The dataset is parsed following the *static* prescription described in 4.2.

## 4.13 Combined models for outlier detection in latent space[21]

First detailed in Ref. [118], this method involves training various anomaly detection methods within the latent space of a variational autoencoder, and then performing combinations of these anomaly scores to determine the optimal method. In the previously referenced paper we show that training an anomaly detection method on latent space representations of events dramatically improves the performance, and that combining these methods allows more information to be extracted. Our process is as follows:

1. Define a VAE architecture.

2. Train it on a subset of the background data.

3. Pass the remainder of background data + signal through the VAE and obtain the latent space representations for each event.

4. Train further anomaly detection algorithms on the latent space representations of the background events (isolation forest (IF), Gaussian mixture model (GMM), static autoencoder (AE), and KMeans)

5. Pass the remaining background and signal events through these algorithms, obtaining 5 measures of anomalousness for each event (VAE reconstruction loss, IF mean path length, GMM log likelihood, AE reconstruction loss, and KMeans distance to nearest centroid).

---

[21]Adam Leinweber, Roberto Ruiz, Melissa van Beekveld, Luc Hendriks, Martin White

Table 22: ALAD model hyper-parameters.

| Parameter | Values |
|---|---|
| batch size | $[1000, 10000]$ |
| $\beta$ term | $[1e{-}5, 1e{-}4, 1e{-}3, 0.01, 0.1]$ |
| latent space dimensions | $[4, 13, 20, 30]$ |

6. Normalise each anomaly score to uniform background efficiency.

7. Perform various combinations (logical and/or, average, and product).

8. Construct a ROC curve and compare the area under the curve (AUC), and signal efficiencies at various background efficiencies.

We have defined numerous variational autoencoders with differing $\beta$ terms, latent space dimension, and batch size in order to determine the optimal configuration.

The reconstruction loss of the VAE consists of four different components: an MSE on the number of objects $x_n$, an MSE on the dimensionless 4-vector terms ($\vec{x}_{r,i} = p_{\mathrm{T}}/\mathrm{GeV}, \eta$ or $\phi$), and a categorical cross-entropy [119] on the categorical variables $x_{c,i}$ that represent different objects in an event (jet, $b$-jet, electron, etc.). The last component is the KL divergence that ensures that the events in the latent space are grouped to a Gaussian. The total loss function of our VAE is then defined as

$$
\begin{aligned}
\mathcal{L} \;=\; & 100\beta \left(x_n - \hat{x}_n\right)^2 \\
& + \frac{\beta}{d_r} \sum_i^{d_r} \left(x_{r,i} - \hat{x}_{r,i}\right)^2 \\
& - \frac{10\beta}{d_c} \sum_i^{d_c} \left(x_{c,i}\log(\hat{x}_{c,i}) + (1-x_{c,i})\log\left(1-\hat{x}_{c,i}\right)\right) \\
& + (1-\beta) \sum_i^{d_z} \mathrm{KL}\left(\mathcal{N}(\hat{\mu}_i, \hat{\sigma}_i), \mathcal{N}(0,1)\right).
\end{aligned}
\tag{25}
$$

Here, $\hat{x}_n$ represents the predicted number of objects, $\hat{x}_{r,i}$ represents the $i$-th predicted regression label, $\hat{x}_{c,i}$ represents the $i$-th predicted categorical label, $d_r$ represents the number of regression variables, and $d_c$ represents the dimensionality of the categorical data. The relative importance of each of these contributions to the loss function is indicated by $\beta$. The anomaly score of an event is given by the reconstruction loss term (the first three lines of Equation 25).

The architecture consists of 3 fully-connected hidden layers for the encoder and decoder, each containing 512, 256 and 128 nodes for the former, and 128, 256 and 512 nodes for the latter. The activation function used between the hidden nodes is the exponential linear unit (ELU) [120]. Tab. 22 contains a summary of the different values of the latent space dimensionalities, batch sizes and $\beta$ terms that are explored in this analysis.

The algorithms trained in the latent space of this VAE (step 4) are an isolation forest [121], a Gaussian mixture model [122], a static autoencoder [123], and a $k$-means clustering algorithm [124]. These algorithms were chosen to utilize a variety of anomaly detection metrics. Each algorithm learns information about the background in a different manner, and as such there is information to be gained by combining the results.

In order to combine our anomaly detection techniques, they must be normalised to uniform background efficiency. For each anomaly score distribution a function $f(x)$ is constructed which returns the background efficiency at a given anomaly score value $x$. Let $g_{\text{bkg}}(x)$ represent the number of background events with anomaly score *greater* than $x$, and $N_{\text{bkg}}$ be the total number of background events. This function $f(x)$ is then given by:

$$f(x) = \frac{g_{\text{bkg}}(x)}{N_{\text{bkg}}} . \tag{26}$$

The signal and background datasets are then normalised by computing $f(x)$ for each signal and background anomaly score. Finally, we construct various combinations. For a given event, let the anomaly score normalised to uniform background efficiency be $x_i$ where $i$ denotes the anomaly score algorithm. The combinations used in this analysis are defined as such:

- AND: $x^{\text{AND}} = \min(x_i)$,

- OR: $x^{\text{OR}} = \max(x_i)$,

- Product: $x^{\text{product}} = \prod_i x_i$,

- Average: $x^{\text{average}} = \frac{1}{N} \sum_i x_i$,

where $N$ is the number of algorithms being used.

## 5    Results[22]

The results of the various anomaly detection methods applied to the physics signals are presented here for each of the figures of merit, i.e. the AUC and the signal efficiencies at a background efficiency of $10^{-2}$, $10^{-3}$, and $10^{-4}$ (see Sec. 2.2.1). First we examine the new physics signals used as benchmarks and then combine the signals to determine which methods are most likely to discover new physics. We mainly show two types of visualization that we will discuss below.

### 5.1    One anomaly algorithm and many signals shown as Box-and-whisker plots

As this study covers many methods tried on many signals, the results will be presented as box-and-whisker plots, as exemplified in Fig. 10. In this plot, the AUC for each signal coming from the *Combined-PROD-VAE_beta1_z21-Flow* (Sec. 4.10) method is denoted by the data points. To summarize these, a box is drawn spanning the inner half of the data. A line through the box marks the median. Whiskers extend from the box to either the maximum and minimum unless these are further away from the edge of the box than 1.5 box lengths. The outlier points are not removed, but are shown as circles. Thus, in this example we see that the *Combined-PROD-VAE_beta1_z21-Flow* method has a very high AUC for most signals, but has a few that perform close to a random guess.

Here, we do not show all figures displaying the performance of each of the anomaly detection algorithms for all signals since the number of different algorithms we tested (including different hyper-parameters) is 1048. The reader can find detailed information on GitHub [125]. We continue this chapter with a discussion of a different type of visualization and then discuss selection criteria for choosing the best performing anomaly detection algorithms.

---

[22]The data for the hackathon results are publicly available at https://github.com/bostdiek/DarkMachines-UnsupervisedChallenge [125]. The website contains easy-to-follow instructions for adding the results for a new model to the figures of this section for subsequent development of improved models.

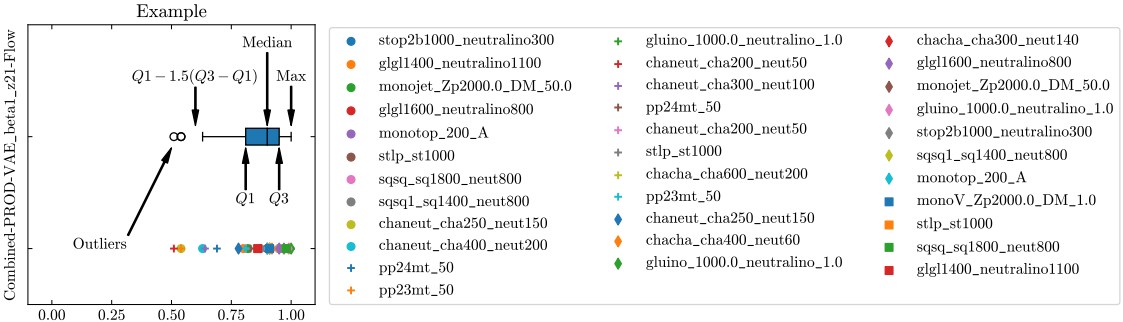

Figure 10: Example of a box-and-whisker plot. The AUC for the *Combined-PROD-VAE_beta1_z21-Flow* method is marked by the data points for each of the 34 channel-signal combinations. This method uses a combination of flow based likelihoods and a variational autoencoder with a loss function only focused on the KL divergence of the latent space (see Sec. 4.10 for more details). A box is drawn spanning the inner half of the data (Qn denotes the nth quartile), with a line through the box at the median. The whiskers extend to the extremal points unless they are further away from the box than 1.5 times the length of the box. These data points are denoted by circles.

## 5.2 All anomaly algorithms and one signal shown as Box-and-whisker plots

One question that we aim to answer in this work is if a good anomaly detection method can discover many models (and ideally any model) of new physics. To help assess this, we show the scores for the individual figures of merit for each signal in Fig. 11. The box-and-whiskers now summarize the over 1000 anomaly detection models. Each row denotes a given new physics signal with the color of the data representing which channel the search is being performed in.

The most important take-away is that some signals are much more difficult for the anomaly detectors than others. For instance, in the chargino-neutralino models with small mass splittings, most of the anomaly detectors have an AUC of around 0.5, equivalent to a random guess. The small mass splittings typically lead to less energetic objects/events, thus, the anomalous events are not in the tails of the distributions and harder to detect. In contrast, the gluino-neutralino signal as well as the RPV stop signal have high AUCs for most detection techniques.

Further study reveals that Channel 2a has consistently worse scores than the other channels. This channel has the tightest pre-selection cuts, yielding the smallest training set. As most of the anomaly detection techniques rely on learning the background well, thus having less data affects the performance. While it would be possible to artificially enhance the training set, this is beyond the scope of this work. The physics signals that only show up in Channel 2a are therefore much more difficult to probe.

The top methods for each signal are available on GitHub.

### 5.2.1 Which algorithm is best? Combining the figures of merit for all physics signals

The results in Fig. 11 indicate that some signals are difficult to discover with anomaly detection techniques. However, we also want to know what techniques work the best for most of the physics signals. With this in mind, we compare the figures of merit (see Sec. 2.2.1) for each anomaly detection technique applied across all of the physics signals. We then look at six different ways of defining 'best'.

Analysis of all models on all signals in the
*Dark Machines Unsupervised Challenge Hackathon Data*

Figure 11: Box plots for each of the physics signals in the hackathon dataset. These summarize the span of results for the many anomaly detection models trained on background only samples. Channel 2a has the tightest pre-selection cuts, and therefore less data, which leads to the signals looking less anomalous.

**A) Top scorer method:** The first method is straightforward: take the models which have the highest score the most number of times. These are summarized in Fig. 12. Each row denotes a given anomaly detection method (including the relevant hyper parameters). The different panels show the four figures of merit. The box plots are colored in according to which models had the top score the most number of times. For instance, the Flow-Efficient Likelihood has the best AUC most often. The *VAE_HouseholderSNF* (Sec. 4.5) yields the highest AUC the 3rd most number of times, and so on. The box plots that are grey are not in the top five techniques for that figure of merit, but are for another figure of merit in the figure.

It is not obvious if having the top score the most times is actually the correct definition of

Best models on all channels of *hackathon data* combined based on top score on a signal

**Figure 12:** Box plots summarizing the anomaly detection techniques applied to all of the new physics signals. The colors denote the technique that have the top score the most times.

'best'. For instance, it is possible that some technique was consistently the second best, with the 'best' models being the best just one time for that particular signal, and performing very badly for the other signals.

**B) Top 5 method:** To assess this, our second method consists of counting the number of times that each technique is within the top 5 scores for each figure of merit. The result is displayed in Fig. 13. While most of these techniques also appeared in the top 1 summary, more of the combination methods show up. Thus they were never the best, but consistently had high scores. Meanwhile the *DeepSVDD* techniques dropped from the list, indicating that they were best on a single physics model, but not as good overall.

Best models on all channels of *hackathon data* combined based on top 5 score on a signal

**Figure 13:** Box plots summarizing the anomaly detection techniques applied to all of the new physics signals. The colors denote the technique that appear in the top five scores the most times.

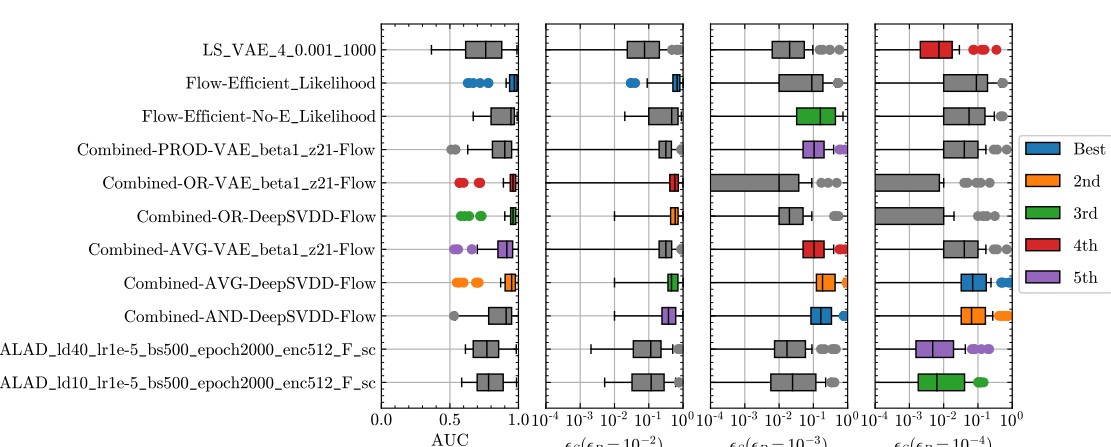

Figure 14: Box plots summarizing the anomaly detection techniques applied to all of the new physics signals. The colors denote the techniques that have the highest average rankings.

**C) Average ranking method:** Counting the number of times that a technique has the top one or one of the top 5 scores is useful in determining the best technique. However, one aspect that is not captured in this is whether a technique does very poorly on a few signals, but very well on others. To get a sense of this, our third method consists of sorting the anomaly detectors based on their average ranking. With this ranking, the model will not have a good ranking if it is the best on one signal and the worst on another. We show the results using this ranking in Fig. 14. We note that the methods combining a fixed target (either *DeepSVDD* or the *VAEs* with $\beta = 1$) and a *Flows* are again among the best techniques. In addition, we see that some standard *VAEs* with small $\beta$ values (i.e. the loss is weighted towards the reconstruction loss) do well for very low background efficiencies, but have not appeared in the previous metrics. We observe that the *ALAD* models perform well at low background efficiency (large background rejection).

**D) Highest mean score method:** Each of the previous three metrics for determining the best technique have been based on the rank ordering of the figures of merit. The methods using a fixed target combined with the flow likelihood have consistently been among the top models. It is also interesting to look at the numerical value of the figure of merit, rather than just the ordering of the results. In Fig. 15 the models we show our fourth method of ordering the algorithms, and use the highest mean scores for each figure of merit.

**E) Highest median score method:** Our fifth method uses the median score, which gives a better sense of the score across the physics signal space. These results are shown in Fig. 16. The median score gives a sense of what methods work best for most new physics signals.

**F) Highest minimum score method:** Another point of interest is to examine the minimum score for each method, which is our sixth and final way to determine which method is the 'best' one. With this, we can find the techniques which have the highest minima. These models are shown in Fig. 17, and there are many interesting aspects. First, the list of the top techniques does not contain any of the combined methods which have otherwise been top methods. The next aspect of note is that the methods which have the highest minimum $\epsilon_S$ for $\epsilon_B = 0.01$ do not score very well on the AUC. The AUC is dominated by large $\epsilon_B$, demonstrating that just

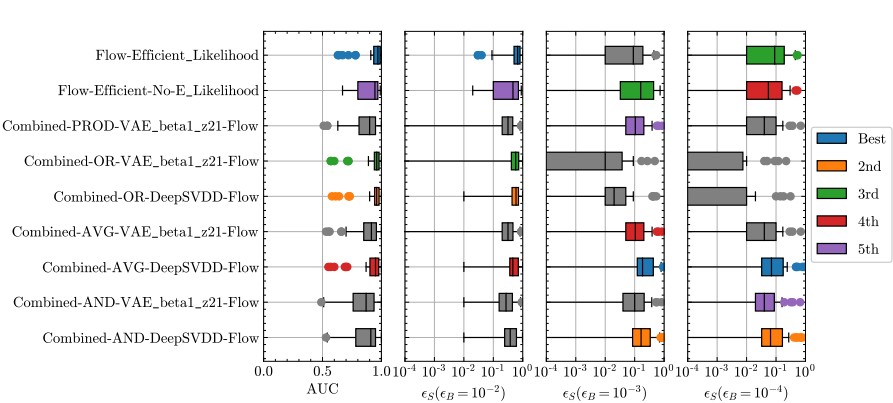

Figure 15: Box plots summarizing the anomaly detection techniques applied to all of the new physics signals. The colors denote the techniques that have the highest mean scores for each of the figures of merit.

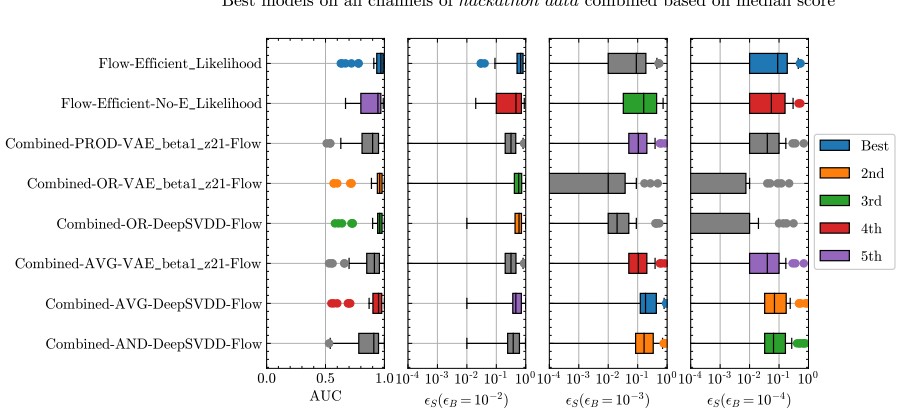

Figure 16: Box plots summarizing the anomaly detection techniques applied to all of the new physics signals. The colors denote the techniques that have the highest median scores for each of the figures of merit.

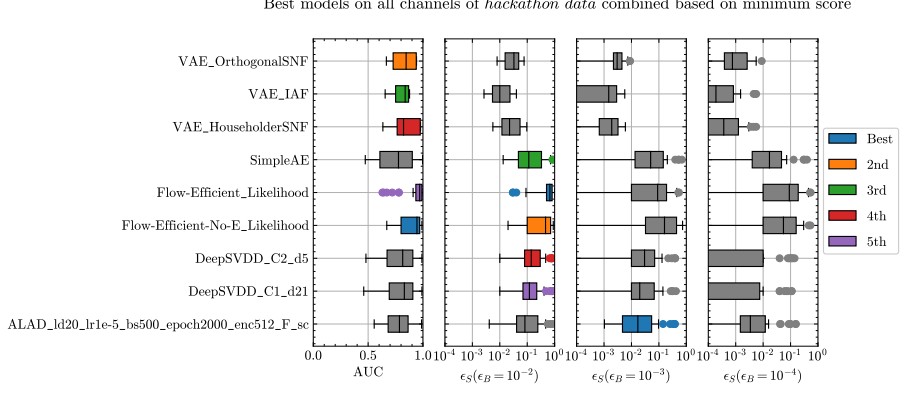

Figure 17: Box plots summarizing the anomaly detection techniques applied to all of the new physics signals. The colors denote the techniques that have the highest minimum scores for each of the figures of merit. No technique has $\epsilon_S$ above 0 for all physics signals for $\epsilon_B = 10^{-4}$ and only one ALAD model has $\epsilon_S$ above 0 for all physics signals at $\epsilon_B = 10^{-3}$.

having a large AUC does not necessarily lead to having a useful classifier for digging deep into the background. The final, and most striking, aspect of the largest minimum scores shown in Fig. 17 is that the $\epsilon_S(\epsilon_B = 10^{-4})$ columns are all gray. This indicates that no method was able to yield a non-zero signal efficiency for *all* physics signals at these tight background cuts. Similarly, the *ALAD* model (see sec. 4.12) is the only one which give a non-zero efficiency for all physics signals tested at a background efficiency at $10^{-3}$. However, these statement sounds more pessimistic than they actually are. Currently, the summaries are being shown for each physics signal in each channel. If a technique works for a given physics model in Channel 2b, but not in 2a (which has the least data), the current analysis uses the minimum from 2a. This motivates a more holistic approach discussed in the next section, looking at the physics models as a whole and looking for the best chance to discover them across channels.

To summarize this section, we have examined six different methods to determine the best anomaly detection techniques. These ranged from sorting the rankings as well as the actual scores across the figures of merit. Often, the methods which were best according to the AUC were sub-optimal when considering the signal efficiency at fixed background efficiency. Further, some methods are better at very tight cuts, but do not work well at the moderate background efficiencies. The techniques which were often in the list of best models include the *Flow-Efficient Likelihood* and *Combined* methods using the Flow Likelihood with either a VAE model with $\beta = 1$ or ensemble of Deep SVDD models (see sec. 4.10).

### 5.2.2 Significance improvement

The chances to discover new physics depend both on the complexity of the signal as well as the cross section for production at the LHC. For this work, we wish to remain agnostic about the cross section. However, we would like to provide a way to combine the different working points into a collective view of how the anomaly detectors are helping.

To keep things simple, we will assume that there are enough events such that the background counts are well modeled by Gaussian statistics. This means that the standard deviation is equal to the square root of the counts. Thus, in a given channel, the significance of the new physics signals can be estimated in terms of the significance before any selection is applied

$$\sigma_S = \frac{S}{\sqrt{B}},\tag{27}$$

where $S$ and $B$ are the number of signal and background events, respectively. When the anomaly detector is applied to the channel, the number of signal and background events changes by $\epsilon_S$ and $\epsilon_B$, leading to the significance after applying the anomaly detection (AD) selection:

$$\sigma_{AD} = \frac{S'}{\sqrt{B'}} = \frac{\epsilon_S\, S}{\sqrt{\epsilon_B\, B}} = \frac{\epsilon_S}{\sqrt{\epsilon_B}} \times \sigma_S.\tag{28}$$

From this, we define the significance improvement (SI) as

$$\mathrm{SI} = \epsilon_S / \sqrt{\epsilon_B}.\tag{29}$$

This metric does not tell us if the anomaly detection technique is capable of discovering new physics, as this still depends on the cross sections, but it informs us on how much the anomaly detector can enhance the statistical purity of the signal over the SM noise

For some methods, this will be for a looser selection ($\epsilon_B = 10^{-2}$), while for others it will be for the tightest background selection ($\epsilon_B = 10^{-4}$). Using the SI, we can turn the figures of merit for each technique into a single number, the maximum SI across the working points. The maximum SI is defined as the maximum significance improvement over over the three working points ($\epsilon_B = 10^{-2}$, $10^{-3}$, and $10^{-4}$).

Analysis of all models on all signals in the
*Dark Machines Unsupervised Challenge Hackathon Data*

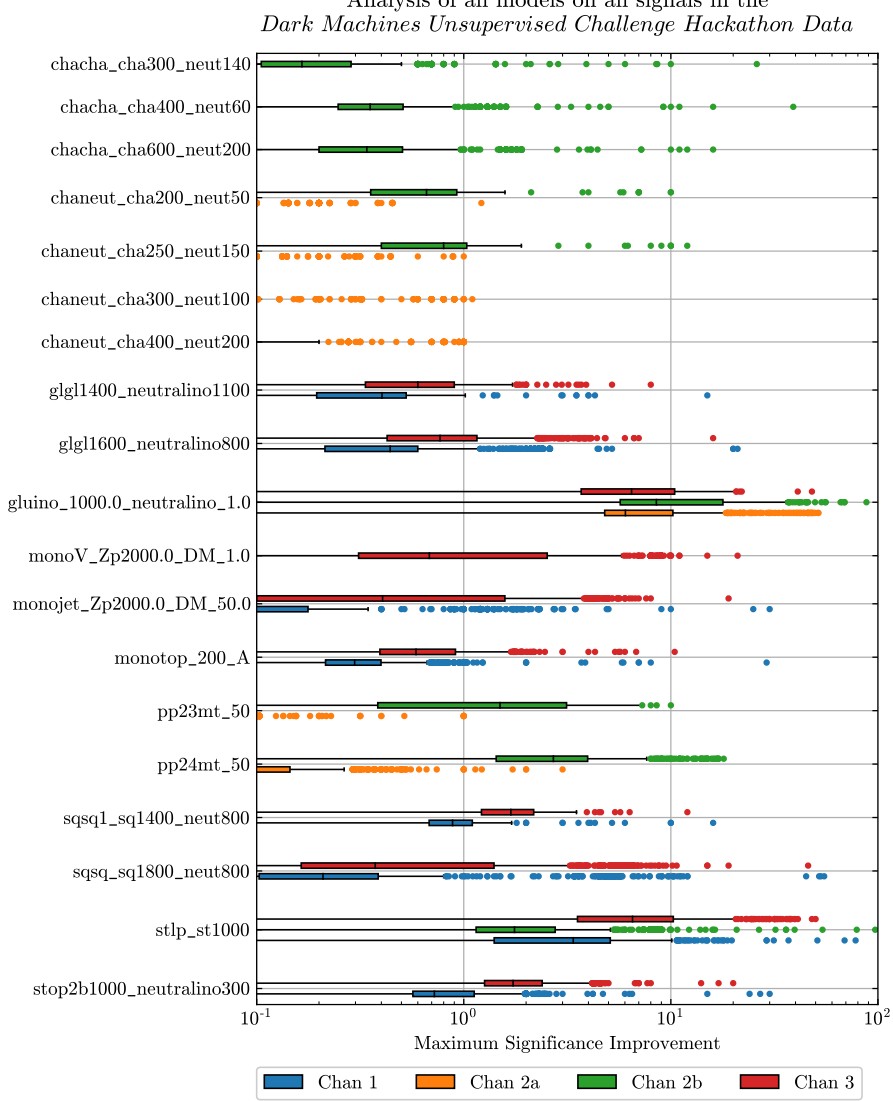

Figure 18: Box plots for each of the physics signals in the hackathon dataset. These summarize the span of results for the many anomaly detection models trained on background only samples. The SI is defined as $\epsilon_S/\sqrt{\epsilon_B}$. The maximum significance improvement over the three working points ($\epsilon_B = 10^{-2}$, $10^{-3}$, and $10^{-4}$) are used as the metric for each technique.

In Fig. 18, we display box plots for the maximum SI for each of the physics models, broken down by channel. From these, it is apparent that the chargino-neutralino signals are very challenging to find using anomaly detection techniques–the best techniques only allow for unit significance improvement. However, we see that as long as a physics signal shows up in any other channel, there is at least one technique that yields has a maximum SI greater than 1.

As a final step, we analyze the maximum SI over the various physics signals for each of the anomaly detection techniques and combine the signals in multiple channels. This means that if a method has a significance improvement of 1.0 for a signal in channel 2a but an improvement of 3.2 in channel 2b, we will only report the number 3.2. With this metric, which we call *total improvement* (TI), we then examine the minimum, median, and maximum scores across each of the physics models.

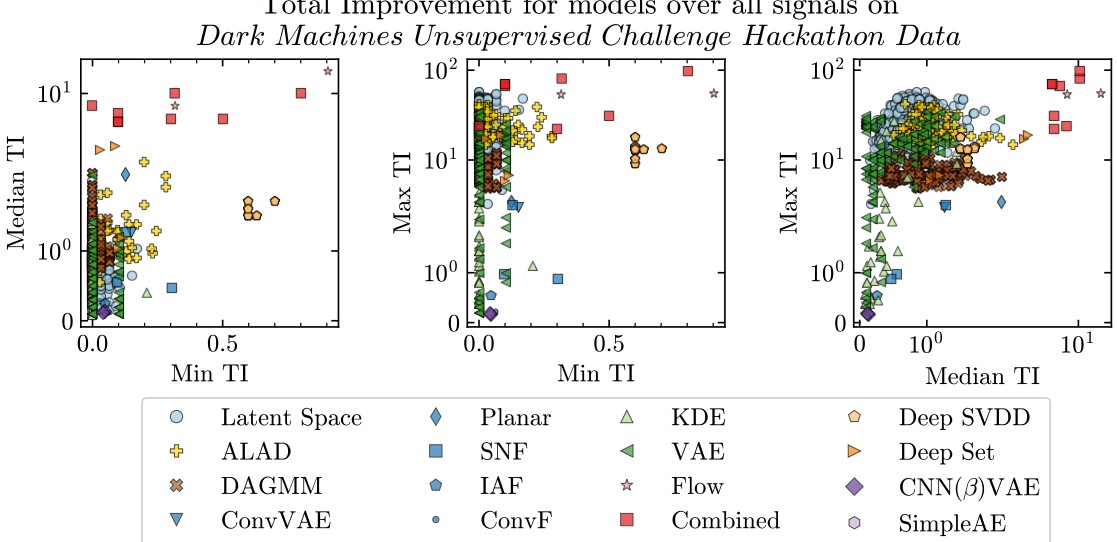

Figure 19: The minimum, median, and maximum best total improvements for each technique across the physics models. The TI is defined as the maximum signal improvement for a physics model across all signal regions.

Fig. 19 shows a scatter plot of the results. The individual data points represent anomaly detection techniques. The colors and shape of the markers denote the family of the technique. The left panel displays the minimum and median TI where two clusters stand out as different from the others. The first is the *Deep SVDD* methods (see sec. 4.9) which have some of the highest minimum significance improvements. However, they have only an average median improvement. Despite having a relatively high minimum, this is still less than unity, implying that using the technique will make it harder to discover some physics models. The other methods to note are the *Flows* (a spline autoregressive flow, sec. 4.8), and the *Combined* techniques (spline autoregressive flow with Deep SVDD or VAE with $\beta = 1$, sec 4.10). These methods offer the largest median significance improvements across the physics models. All of the other techniques are clustered towards smaller median and minimum significance improvements.

The second and third panels show the maximum TI along the $y$ axis. Looking back to Fig. 18, we see that the maximum significance improvement is dominated by the score on the easy to find gluino-neutralino or RPV-violating stop models. The methods doing anomaly detection inside the *latent space* (sec. 4.13), as well as the *ALAD* models (sec. 4.12), have large maximum significance improvements, even though their medians are rather small.

We plot the methods in three dimensions in Fig. 20 to obtain a clearer picture of the results. This shows clearly that the *Deep SVDD*, *Flows*, *Deep Sets* (sec. 4.3), and *Combined* methods stand out from the rest of the methods. We expect that the methods with the highest median improvement will perform best on the blinded dataset. For this reason, we select all of the models which have a median TI greater than 2 to be passed on the the blinded dataset. The selection of models passing this cut are shown in Tab. 23.

The method with the largest median significance improvement is the *Flow-Efficient Likelihood*, one of the spline autoregressive flows of sec. 4.8. The next highest medians also have larger maxima, and are both *combined* methods using the flow likelihood with an ensemble of *Deep SVDD* (sec. 4.10). It is interesting that many of the best methods use a constant target value in the loss function. This means they do not do any reconstruction and only concern compressing the input to a constant. With no attempt at decoding, it may be more challenging to validate for use at the LHC. Further study of this topic is required.

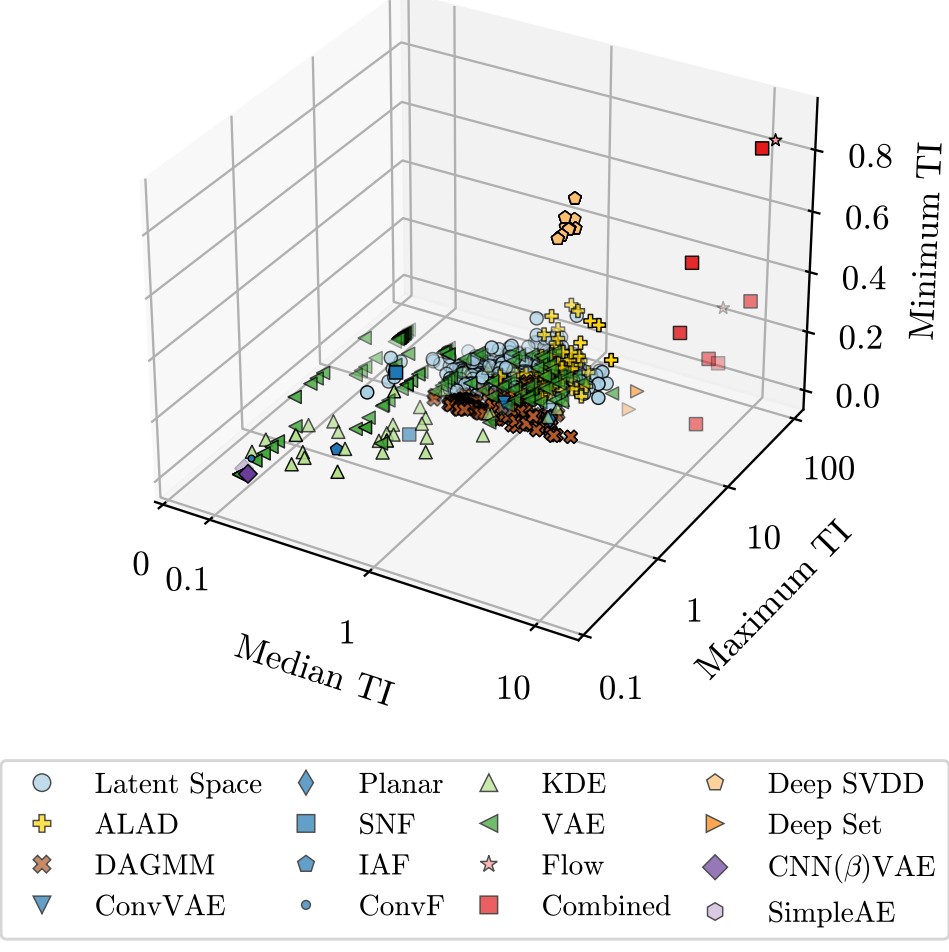

Figure 20: The minimum, median, and maximum best total improvements for each technique across the physics models.

## 5.3 Results: The secret and blinded darkmachines dataset

We now discuss the results of the various anomaly detection methods applied to the blind darkmachines dataset. For this, we only use the subset of the models that performed best in the previous section. Since we do not know what kind of signal to find in this data challenge, we use a metric that infers the ability to find "discoverable" signal models. Some signal models are not found well by any algorithm. We will use the median TI metric to account for this, as selecting on the minimum or maximum TI might bias our results. Therefore, the anomaly classifiers with a median TI greater than 2 are selected to participate in an unlabeled data hackathon set (the blind darkmachines dataset). To allow for comparison with the other techniques, we have also included some of the other algorithms. A full list is shown in Tab. 23 with their minimum, median and maximum TI on the hackathon dataset.

Firstly, we show the box plots for the average ranked, medium score, and minimum score of the figures of merit described in Sec. 2.2.1 in Fig. 21. We combine all channels for these results. We see that the *Combined* algorithms (Sec. 4.10) and also the SimpleAE (Sec. 4.1) consistently performs well. In Sec. 5.2.1 we found that the *Combined* algorithms performed well when using the average rank metric, but also the *Flow-Efficient_Likelihood* algorithm performed well there. On the Secret dataset, we find that this algorithms does not make it to the top 5 for

Table 23: Selection of models from the Hackathon dataset that are chosen to be applied to the Secret dataset. The TI scores represent those from the Hackathon challenge. The first block of models have a median TI greater than 2, and are expected to generalize well to unknown new physics. The second block of models were chosen for comparison.

| Name | Min TI | Median TI | Max TI | Section |
|---|---|---|---|---|
| Flow-Efficient_Likelihood | 0.90 | 15.00 | 54.00 | 4.8 |
| Combined-AND-DeepSVDD-Flow | 0.32 | 10.00 | 79.00 | 4.10 |
| Combined-AVG-DeepSVDD-Flow | 0.80 | 10.00 | 97.00 | 4.10 |
| Flow-Efficient-No-E_Likelihood | 0.32 | 8.00 | 53.00 | 4.8 |
| Combined-PROD-DeepSVDD-Flow | 0.00 | 7.91 | 24.00 | 4.10 |
| Combined-AND-VAE_beta1_z21-Flow | 0.10 | 7.00 | 66.00 | 4.10 |
| Combined-OR-DeepSVDD-Flow | 0.50 | 6.30 | 31.00 | 4.10 |
| Combined-OR-VAE_beta1_z21-Flow | 0.30 | 6.30 | 22.00 | 4.10 |
| Combined-AVG-VAE_beta1_z21-Flow | 0.10 | 6.00 | 69.00 | 4.10 |
| Combined-PROD-VAE_beta1_z21-Flow | 0.10 | 6.00 | 69.00 | 4.10 |
| DeepSetVAE_beta_1.0_weight_10.0 | 0.08 | 3.75 | 18.81 | 4.3 |
| DeepSetVAE_beta_1.0_weight_1.0 | 0.03 | 3.50 | 16.91 | 4.3 |
| ALAD_ld10_lr1e-5_bs500_epoch2000_enc512_F_sc | 0.19 | 2.85 | 14.63 | 4.12 |
| DAGMM_10_1e-07_1000_0.01_0.01_50_15_8 | 0.00 | 2.31 | 6.51 | 4.11 |
| DAGMM_10_1e-07_1000_0.001_0.01_50_15_8 | 0.00 | 2.31 | 6.51 | 4.11 |
| ConvVAE_PlanarFlow (Planar) | 0.12 | 2.28 | 3.45 | 4.5.1 |
| VAE-dynamic-beta1-z13_Radius | 0.00 | 2.20 | 28.00 | 4.2 |
| ALAD_ld10_lr1e-5_bs5000_epoch2000_enc512_F_sc | 0.28 | 2.20 | 17.40 | 4.12 |
| KDE_PCA_D=9 | 0.00 | 1.43 | 12.25 | 4.7 |
| ALAD_ld10_lr1e-5_bs5000_epoch2000_enc512_L_1 | 0.00 | 1.40 | 24.27 | 4.12 |
| ALAD_ld10_lr1e-5_bs5000_epoch2000_enc512_L_2 | 0.00 | 1.21 | 20.73 | 4.12 |
| ALAD_ld10_lr1e-5_bs5000_epoch2000_enc512_L_sc | 0.00 | 1.06 | 36.94 | 4.12 |
| ALAD_ld10_lr1e-5_bs500_epoch2000_enc512_L_1 | 0.03 | 1.06 | 28.27 | 4.12 |
| ALAD_ld10_lr1e-5_bs500_epoch2000_enc512_L_sc | 0.22 | 1.04 | 38.17 | 4.12 |
| SimpleAE | 0.20 | 0.98 | 39.60 | 4.1 |
| ALAD_ld10_lr1e-5_bs500_epoch2000_enc512_L_2 | 0.17 | 0.91 | 21.49 | 4.12 |
| ConvVAE_ConvolutionalFlow (ConvF) | 0.06 | 0.12 | 0.21 | 4.5.4 |

neither of the figures of merits in the average rank. A similar conclusion holds for the median score metric. On the other hand, the *DeepSetVAE* and the *SimpleAE* perform better on the Secret dataset than on the Hackathon datasets using these two metrics. For the miminum score metric, we see that the *ALAD*, *SimpleAE* and the *Flow-Efficient* methods work well, which was also observed in the Hackathon dataset. The *Combined* methods perform less well using this metric. For all algorithms we find that there are some signals where they perform worse than a random guess, which we also show in these plots for comparison.

We now move on to the TI merit to asses which model performs best on the Secret dataset. The minimum, median and maximum TI scores are shown on a 3D projection in Fig. 22. The highest minimum TI is found for the *SimpleAE* algorithm. The *Flow-Efficient* and one of the *Combined* methods also show a relatively high minimum TI. This *Combined* algorithm also

Best models on all channels of secret dataset combined based on average rank

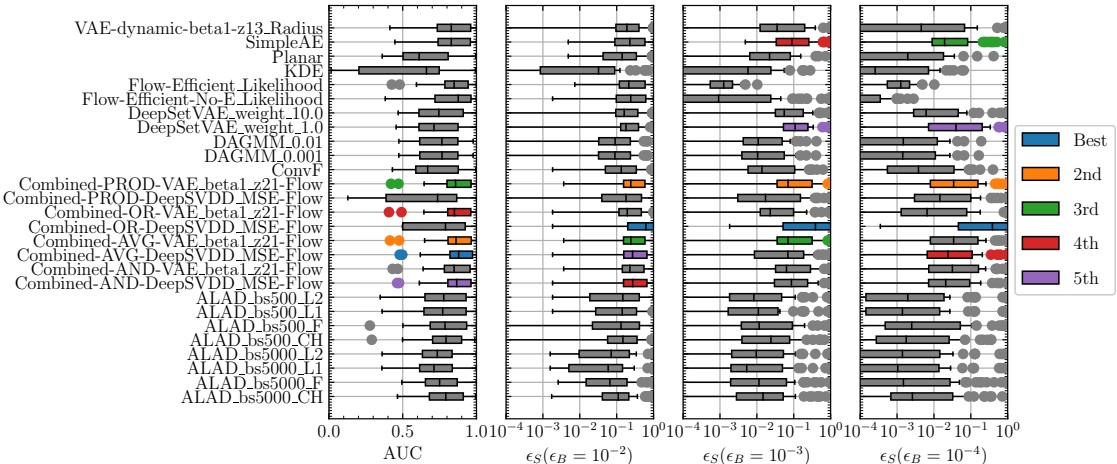

Best models on all channels of secret dataset combined based on median score

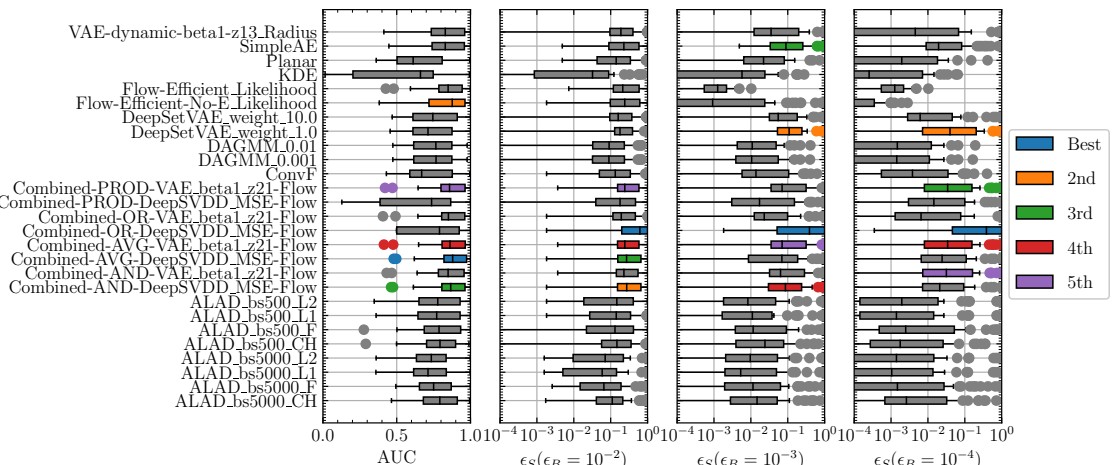

Best models on all channels of secret dataset combined based on minimum score

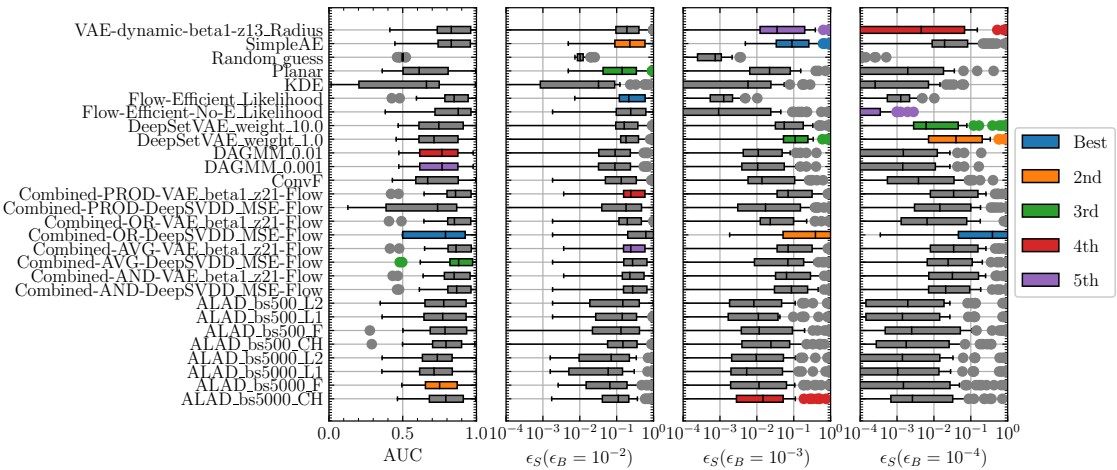

Figure 21: Box plots summarizing the anomaly detection techniques applied to the secret dataset. The colors denote the techniques that have the highest average rank (top), median score (middle) and minimum score (bottom) for each of the figures of merit described in Sec. 2.2.1.

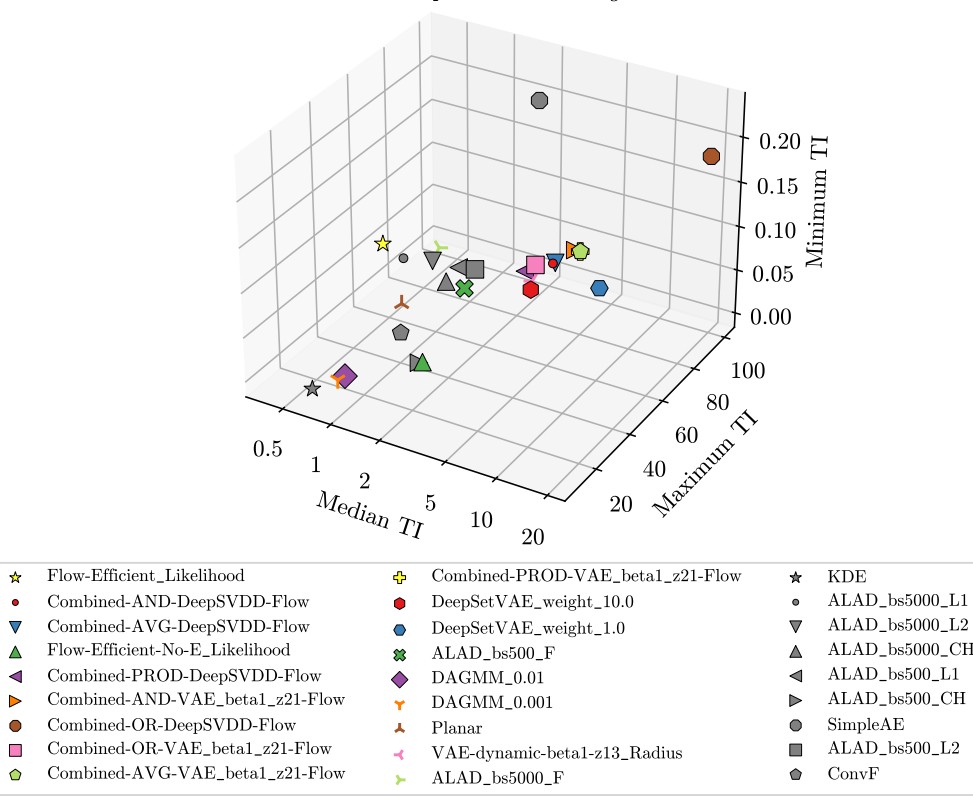

*Dark Machines Unsupervised Challenge Secret Data*

Figure 22: The minimum, median, and maximum best total improvements for each technique applied on each of the signals in the secret dataset.

shows a significantly higher median TI score than the other algorithms. The other versions of the *Combined* algorithm show persistently high scores for the maximum TI, but perform less well for the median and minimum TI metrics. This indicates that these algorithms are generally good at detecting one specific signal, but not at detecting new physics in general. The same can be said for the *ALAD* methods, and the *DeepSetVAE*. The *KDE* (Sec. 4.7) and the *DAGMM* methods (Sec. 4.11) seem to perform badly on all three figures of merit.

It is interesting to observe that the relative performance of the algorithms on the Hackathon dataset does not directly reflect their relative performance on the Secret dataset. To get a better sense of this, we show the 2D projections of the TI scores for the selected models for the Hackathon dataset (upper panel) and the Secret dataset (middle panel) in Fig. 23. The general trend is that the minimum TI is lower (worse) for the Secret dataset than for the Hackathon dataset. This implies that one of the unknown signals is much harder to find with anomaly detection technique. Another similar trend is that most models have higher maximum TI on the Secret dataset than on the Hackathon dataset, implying that one of the secret signals is much easier to find than those seen in the Hackathon. This emphasizes the potential downside of the minimum or maximum TI, they are determined by the performance of a single signal.

Ideally, the median TI should reflect the potential for a model to discover a new physics signal. However, this metric does also depend on the ensemble of new physics signals. In particular, we see that many of the methods have lower median TI scores on the Secret dataset than on the Hackathon dataset. Beyond the general trend, there are a few noteworthy examples. On the Hackathon dataset, the *Flow-Efficient_Likelihood* algorithm had the highest median TI, whereas the highest median TI of the Secret dataset is found for the *Combined-OR-*

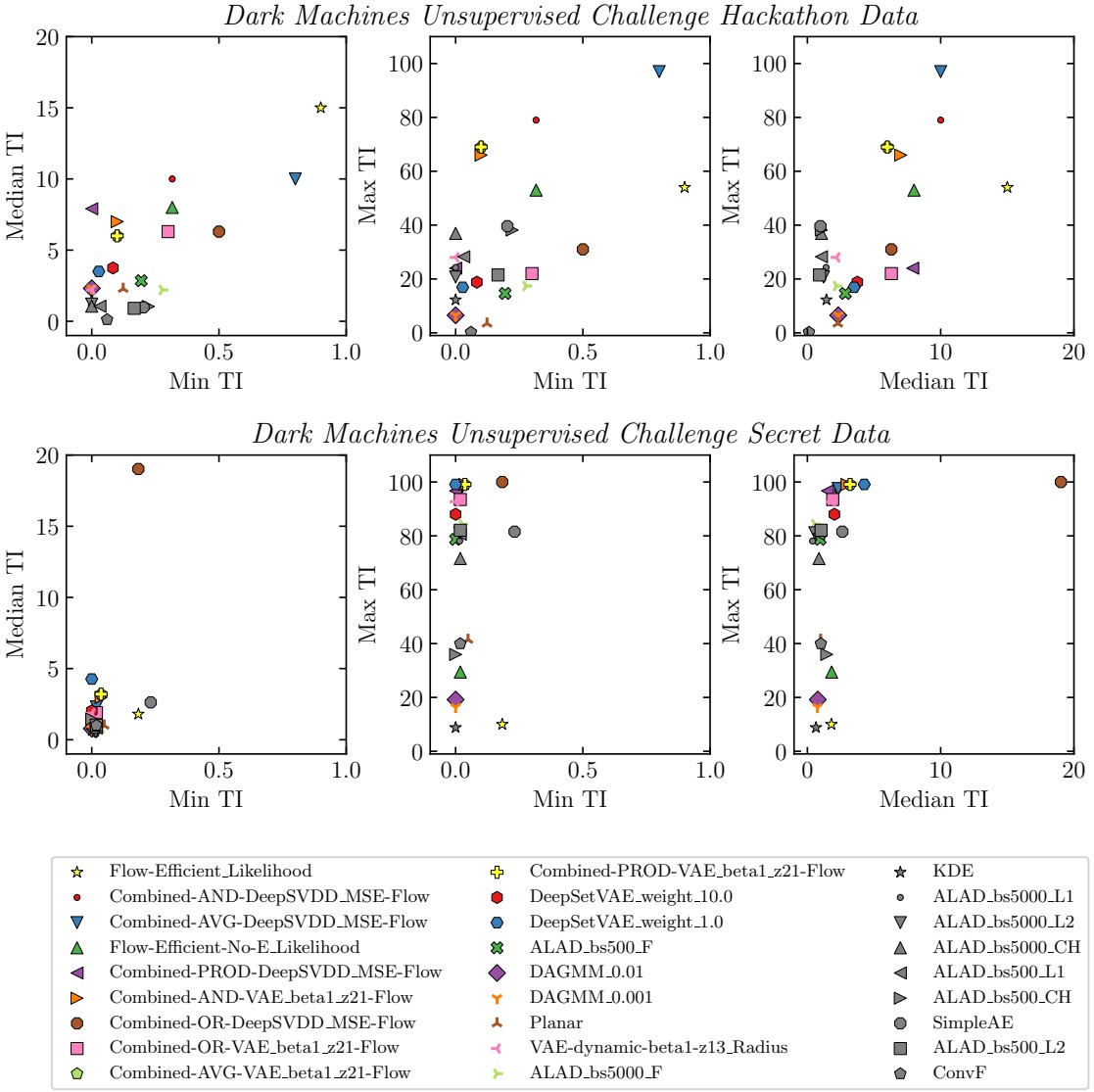

Figure 23: The minimum, median, and maximum best total improvements for each technique applied on each of the signals in the Hackathon (top) and Secret (middle) dataset.

*DeepSVDD_MSE-Flow* method. The *Flow-Efficient_Likelihood* even performs quite poorly on the median TI metric in the Secret dataset.

In Fig. 24, we show the median TI scores for for the selection of models for both the Hackathon dataset (along the *x*-axis) and the Secret dataset (along the *y*-axis). In this plot, it is easy to see that many of the best models on the Hackathon dataset (further to the right of the figure) are no longer the top models for the Secret dataset (further to the top of the figure). However, it is important to note that most of the models that were selected as having high median TI scores on the Hackathon dataset do better than most of the other reference models on the Secret dataset. For instance, the top 5 models on the Secret dataset all had a median TI > 2 on the Hackathon dataset. Additionally 13 of the top 14 models on the Secret dataset had a median TI > 2 on the Hackathon dataset. The *SimpleAE* model had a low TI on the Hackathon dataset but performs relatively well on the Secret dataset. The models with $TI > 2$ on both datasets are shown in Tab. 24.

While the *Flow-Efficient* models still have decent median TI scores on the Secret dataset, it

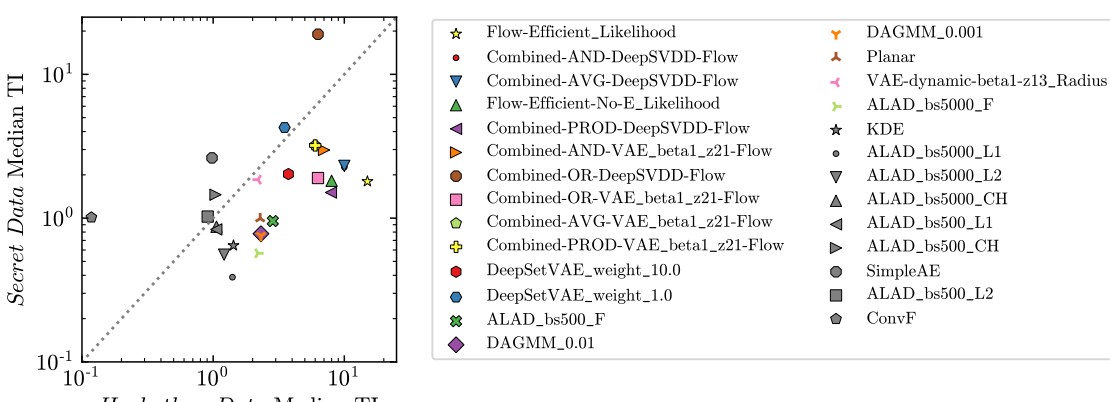

Figure 24: The median TI scores for the Hackathon and Secret datasets along the *x*- and *y*-axis, respectively. The point marked with color were chosen as having median TI > 2 on the Hackathon data, and the grey points are included as reference models.

was surprising to see their scores fall so much. It is unclear what the exact cause is. Possibilities include that there is something underlying the modeling which makes certain signals more susceptible to the anomaly score. Over-optimization seems unlikely as there was no large hyperparameter scan performed. However, there does seem to be some evidence of this when examining the results of the other models. For example, the *ALAD_F* and *DAGMM* models are also well below the diagonal of Fig. 24, while Tab. 5 shows that these chosen models with high median TI on the Hackathon dataset were part of a hyper-parameter scan of 96 models (ALAD) and 384 models (DAGMM). It is important to keep in mind that optimizing for a validation set may still not generalize to an unknown test set.

Table 24: The models which have a median TI score greater than 2.0 on both the Hackathon and Secret datasets. The values show the median TI scores on the respective datasets.

| Model | Hackathon Data | Secret Data |
|---|---|---|
| Combined-OR-DeepSVDD-Flow | 6.30 | 19.02 |
| DeepSetVAE_weight_1.0 | 3.50 | 4.27 |
| Combined-AVG-VAE_beta1_z21-Flow | 6.00 | 3.21 |
| Combined-PROD-VAE_beta1_z21-Flow | 6.00 | 3.20 |
| Combined-AND-VAE_beta1_z21-Flow | 7.00 | 2.98 |
| Combined-AVG-DeepSVDD-Flow | 10.00 | 2.31 |
| Combined-AND-DeepSVDD-Flow | 10.00 | 2.26 |
| DeepSetVAE_weight_10.0 | 3.75 | 2.03 |

# 6 Conclusions

In this paper, we have described benchmark datasets for future studies of new physics detection at the LHC. We described the details of the data generation and the data format, which

allow the user to easily handle the data in any programming language. This data is divided in two different sets in this paper: the Unsupervised Hackathon dataset, used for training and testing of the machine-learning algorithms, and a still-blinded Secret dataset, used to assess the relative performance of the algorithms. This Secret dataset will remain blinded in order to facilitate an unbiased benchmark of future algorithms. Combined, the datasets consist of more than 1 billion simulated LHC SM events, distributed over 4 different analysis channels. It contains more than 40 potential LHC signal samples. The datasets and their Zenodo we-blinks are summarized in Tab. 4 on page 14, and we encourage the community to familiarize themselves with this dataset.

These datasets are used to perform a comparison of 16 different machine-learning methods. More than 1000 variations and combinations of these methods (see Tab. 5) are tested on their ability to determine an anomaly score of LHC events. We propose in Sec. 1 on page 5 how such an anomaly score could be implemented in almost all LHC searches to define model-independent signal regions. The precise definition of the anomaly score depends on the employed algorithm, but as a figure of merit we have used the AUC and the signal efficiencies at a background efficiency of $10^{-2}$, $10^{-3}$ and $10^{-4}$. In addition, as four different channels and $\mathcal{O}(40)$ different signals are involved in both the Secret and Hackathon datasets, we have derived combined figures to derive the mean, maximum and minimum performance scores of the algorithms (see Sec. 5). The results of all of the models presented are available at [125, 126], allowing for an easy comparison with future methods.

We encourage the community to develop new anomaly detection methods using these datasets. However, caution should be used in order to ensure that one is not over optimizing to the validation Hackathon dataset signals. For instance, the methods which employed large hyper-parameter scans were able to find a few models which performed well on the Hackathon dataset. However, these models did not generalize as well to the Secret dataset.

Of the over 1000 submitted models, eight had median TI scores greater than 2.0 for both the Hackathon and the Secret datasets separately. These are shown in Tab. 24 and consist of *Combined* models (Sec. 4.10) and *DeepSet* models (Sec. 4.3). The *Combined* models use a flow based likelihood in combination with either a Deep SVDD model or a $\beta$VAE model with $\beta = 1$. In a Deep SVDD model, the inputs are mapped to a vector of constant numbers. In a $\beta$VAE, using $\beta = 1$ places all of the weight of the loss function on the KL divergence term, so the network maps the inputs to a Gaussian latent space, but does not try to reconstruct the inputs. The particular *DeepSet* models with high scores also used $\beta = 1$ in the loss function. It is interesting that all of top models use some sort of fixed target, either only focusing on the KL divergence of the latent space, or having a component that is mapping the inputs to a fixed number. The reason that these methods seem to generalize better is unknown and left for further study.

# Acknowledgements

MvB acknowledges support from the Science and Technology Facilities Council (grant number ST/T000864/1). The work of A.J. is supported by the National Research Foundation of Korea, Grant No. NRF-2019R1A2C1009419. The work of J.M. is supported in part by the Generalitat Valenciana (GV) through the contract APOSTD/2019/165, and by Spanish and European funds under the project PGC2018-094856-B-I00 (MCIU/AEI/FEDER, EU). The work of R.RdA. and J.M. is supported by the Artemisa project, co-funded by the European Union through the 2014-2020 FEDER Operative Programme of Comunitat Valenciana, project ID-IFEDER/2018/048. The work of J.H. and B.R. is supported by the Royal Society under grant URF\R1\191524. The work of C.D. is supported by the Swedish Research Council. Research

by A.B. is supported by the US Department of Energy under contract DE-SC0011726. The work of B.O. is supported by the U.S. Department of Energy under contract DE-SC0013607. This work is supported by the National Science Foundation under Cooperative Agreement PHY-2019786 (The NSF AI Institute for Artificial Intelligence and Fundamental Interactions, http://iaifi.org/). The work of P.J. was supported by the DIANA-HEP Graduate Fellowship program and the ThinkSwiss Research Scholarship. P.J. M.P, M.T., and K.A.W are supported by the European Research Council (ERC) under the European Union's Horizon 2020 research and innovation program (grant agreement n⁰ 772369. J.M.D. is supported by the DOE, Office of Science, Office of High Energy Physics Early Career Research program under Award No. DE-SC0021187 and by the DOE, Office of Advanced Scientific Computing Research under Award No. DE-SC0021396 (FAIR4HEP). J-R. V. is supported by the U.S. DOE, Office of Science, Office of High Energy Physics under Award No. DE-SC0011925, DE-SC0019227, and DE-AC02-07CH11359. Computations in this paper were run on the FASRC Cannon cluster supported by the FAS Division of Science Research Computing Group at Harvard University. M.W. and A.L. are supported by the Australian Research Council Discovery Project DP180102209 and the ARC Centre of Excellence in Dark Matter Particle Physics (CE200100008). RV acknowledges support from the European Research Council (ERC) under the European Unions Horizon 2020 research and innovation programme (grant agreement No. 788223, PanScales), and from the Science and Technology Facilities Council (STFC) under the grant ST/P000274/1.

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
