# Peer review of "The Dark Machines Anomaly Score Challenge: Benchmark Data and Model Independent Event Classification for the Large Hadron Collider"

_SciPost Physics, doi:SciPost Phys. 12, 043 (2022)_

## Round 1 · Referee Report · Anonymous (Referee 1) · 2021-7-15

Strengths

1- The described technique is, as the authors say "a rather small modification of existing search strategies" which in turn allows existing tools for background determination and significance determination to be re-used (but see caveat under 'weaknesses')

2- It describes a method to discover new physics without relying on properties of a pre-existing model.

3-The multiple definitions of "best" capture the variations between algorithms and let the reader judge for herself the various pros and cons. This is reasonable. Although developed in another context, Arrow's Theorem ensures that there is not a unique definition of "best". Six possibilities are more than I would have expected.

Weaknesses

1- Suppose you find an anomaly. How do you characterize its significance? While the authors claim that this can be addressed using the usual tools for significance determination, it then becomes an a posteriori search. At a minimum a trials factor (sometimes called the look-elsewhere effect) needs to be included in the determination of the significance. Determining the trials factor is far from simple.

I see this as an important hurdle to be overcome before this technique can be used in real data analysis (Certainly one can adopt the strategy of splitting into two independent datasets, as discussed on Page 3, but this increases the statistical uncertainty because only a subset of the data can be used for significance calculation). I don't think it should stand in the way of publication, as this is a step in the right direction.

Report

This paper does two things - it describes events created for the Dark Machines initiative challenge, and it discusses a particular approach to anomaly identification. It would probably have been better to separate them. Criticisms of one part likely do not flow to the other, and the reader may wonder what if any advantages the anomaly identification algorithm has by being written by the same people who created the dataset it was tested on.
If this is my misunderstanding, page 5 needs to be re-written. If this is a correct understanding, https://www.phenoMLdata.org should be updated to reference this paper on the arXiv.

Editorial Suggestions:

While it's nice that you name the developers of various algorithms in footnotes, a reference would be more useful. If this is unique work that is presented for the first time in this paper, shouldn't these people be co-authors (some but not all pf them are)? In any event, the relationship between their work and your work is difficult to find, and finding more information is even more difficult.

P2 Paragraph 3 "To date, no signal of BSM physics has been found at the LHC". I think you want to be specific in what you mean. Certainly, there are signals - at low significance. There was a diphoton bump that went away a few years ago. There are rumors of peculiarities in 4-muon events. There are even a number of relatively high-significance anomalies, especially in the flavour sector explored by LHCb. So what exactly do you mean? That no signal has reached 5 sigma? That no signal in high pT physics has reached 5 sigma? You should be more explicit here.
P3 Paragraph 2 - "unprecedented". Really? Nothing has ever made this much progress in all of history? I think you need to either soften that statement or defend it.
P5 footnote (and page 13) - the document itself should include how to make various requests of the authors, and not rely on the reader fishing for it on the arXiv. Adding an email address or two would be fine.
P6 paragraph 2: The algorithm's name is anti-k_t with a lowercase t.
Figure 6 would be more beautiful if the number of leptons was centered on the bin and not at its left edge.
Page 17 Paragraph 3 - "highly dimensional". I don't know what you are trying to say here. Perhaps "high-dimensional"?

Requested changes

1- See editorial suggestions above 2- Consider referencing instead of footnoting the algorithms used 3- Consider commenting on the global significance issue and the implicit trials factor

  • validity: high
  • significance: good
  • originality: good
  • clarity: high
  • formatting: excellent
  • grammar: excellent

Author:  Bryan Ostdiek  on 2021-10-08  [id 1825]

(in reply to Report 1 on 2021-07-15)
Category:
answer to question
correction

The referee writes:

Suppose you find an anomaly. How do you characterize its significance? While the authors claim that this can be addressed using the usual tools for significance determination, it then becomes an a posteriori search. At a minimum, a trials factor (sometimes called the look-elsewhere effect) needs to be included in the determination of the significance. Determining the trials factor is far from simple. I see this as an important hurdle to be overcome before this technique can be used in real data analysis (Certainly one can adopt the strategy of splitting into two independent datasets, as discussed on Page 3, but this increases the statistical uncertainty because only a subset of the data can be used for significance calculation). I don't think it should stand in the way of publication, as this is a step in the right direction.

Our response: Any of the presented algorithms is not per se an analysis tool that outputs a significance. They could be used as part of a traditional analysis (e.g., to define a signal region) that would do that (which was not in the scope of this paper) or used just to select rare events for visual inspection, from which one could then motivate searches in the future (that would be able to assess a significance). The latter is basically what is now done with model generic searches at the LHC, whose ultimate scope is to identify interesting regions for future searches and not to output a significance.

The current work focuses on testing the sensitivity of different anomaly detection strategies to a signal, and not to the present the whole analysis that would exploit them.

However, one could also put this into a full analysis (and get a significance) by: 1. Define the channel (with precuts) 2. Train an algorithm defining an anomaly score 3. Define Signal Regions with the anomaly score (with 10-3 etc background events) 4. Define control regions as in many other searches 5. Predict the background with a background only fit 6. Determine a p-value / significance with the data and the background prediction in your signal region In this way, there are no trial factors (or only as many trial factors as there are signal regions that are defined).

The referee writes:

The reader may wonder what if any advantages the anomaly identification algorithm has by being written by the same people who created the dataset it was tested on. If this is my misunderstanding, page 5 needs to be re-written. If this is a correct understanding, https://www.phenoMLdata.org should be updated to reference this paper on the arXiv.

Our response: There was a factorization among the authors of the paper. The teams in control of the data generation (especially the secret data) did not submit algorithms for the challenge.

The referee writes:

While it's nice that you name the developers of various algorithms in footnotes, a reference would be more useful. If this is unique work that is presented for the first time in this paper, shouldn't these people be co-authors (some but not all of them are)? In any event, the relationship between their work and your work is difficult to find, and finding more information is even more difficult.

Our response: The algorithms submitted are a mix of original and already existing in the literature. All of the authors of the algorithms (in the footnotes) are also authors of the paper. The editors of the manuscript encouraged the teams with original algorithms to write these up in individual papers, but only a few of the teams have done so.

The referee writes:

P2 Paragraph 3 "To date, no signal of BSM physics has been found at the LHC". I think you want to be specific in what you mean. Certainly, there are signals - at low significance. There was a diphoton bump that went away a few years ago. There are rumors of peculiarities in 4-muon events. There are even a number of relatively high-significance anomalies, especially in the flavor sector explored by LHCb. So what exactly do you mean? That no signal has reached 5 sigmas? That no signal in high pT physics has reached 5 sigmas? You should be more explicit here. Our response: Thank you for this suggestion, we did not realize that there would be confusion about this statement. We have revised it to say, “To date, no BSM physics has been discovered (5 $\sigma$) at the LHC.”

The referee writes:

P3 Paragraph 2 - "unprecedented". Really? Nothing has ever made this much progress in all of history? I think you need to either soften that statement or defend it.

Our response: We have changed “unprecedented” to “significant step forward.”

The referee writes:

P5 footnote (and page 13) - the document itself should include how to make various requests of the authors, and not rely on the reader fishing for it on the arXiv. Adding an email address or two would be fine.

Our response: Thank you for the suggestion, we have added email addresses for the contact persons (under the abstract).

The referee writes:

P6 paragraph 2: The algorithm's name is anti-k_t with a lowercase t.

Our response: This has been fixed.

The referee writes:

Figure 6 would be more beautiful if the number of leptons was centered on the bin and not at its left edge.

Our response: Good suggestion, that has been addressed.

The referee writes:

Page 17 Paragraph 3 - "highly dimensional". I don't know what you are trying to say here. Perhaps "high-dimensional"?

Our response: Thanks for finding this typo.

The referee writes:

Consider referencing instead of footnoting the algorithms used

Our response: Citations have been added for the algorithms which have been written up since.

The referee writes:

Consider commenting on the global significance issue and the implicit trials factor

Our response: As mentioned above, we believe there are ways to implement a search without few trial factors. We have commented on this on pages 3 and 5.

---

## Round 1 · Referee Report · Anonymous (Referee 2) · 2021-7-21

Strengths

1 The paper details several novel, model-independent approaches to search for signals of beyond the standard model physics at the LHC. 2. The chosen methods are state-of-the-art in the field of machine learning and thus the paper encourages their use in the future. 3. A data challenge is described inviting readers to try out similar techniques and links to relevant data samples and analysis codes are given as well.

Weaknesses

  1. Several new machine-learning methods are described in the paper that have been tried out during a data challenge to find new physics signals . However, no conclusive study is given why certain methods outperform others, which was likely beyond the scope of the challenge itself.

Report

I recommend the publication given the novelty of the described approaches for performing searches for new physics at the LHC. The paper is written in a well structured and clear way. It gives sufficient details and cites the relevant literature. Information on reproducing the results are also provided.

Requested changes

  1. There are some typos (e.g. repeated words) and a few suboptimal descriptions (e.g. last 3 sentences in 5.2.2), which should be fixed.
  2. Figure 1 shows the same histograms, which may confuse a reader since it implies that the anomaly score is or could be equivalent to a supervised classifier.
  3. Various phrases seem to be used as synonyms (e.g. signal region <-> signal area; simulated <-> computer-generated; anomaly values <-> anomaly scores; etc.), which can be confusing. It would be best to use a consistent scheme throughout the paper.
  4. It would be good to give a bit more details on the analysis objects: i.e. from what objects are jets clustered (e.g. calorimeter cells)?; how are b-jets identified? are leptons/photons required to be isolated? how are ETmiss and its angle defined?
  5. How have the investigated signal scenarios been chosen, e.g. are these already searched for and if yes point out if the chosen signals have not yet been excluded in direct searches yet.
  6. The chosen channels lack a bit of motivation; e.g. what kind of models can be targeted in each channel?
  7. The shown distributions in Fig. 9 suggests that outliers may always be found in the tail of a distribution. Such events can be easily found as well through a series of selections as in traditional searches. It might instead be worth highlighting that most methods described in the paper could also detect outliers that fall within "voids" in n-dimensions.
  8. A brief discussion should be added to the paper on how to deal with detector noise that might be identified as anomalous as well and can occur at a higher rate than new physics.
  9. The autoencoder described in Section 4.1 performs better in Channel 2a after an optimization of the hyperparameters. It is however surprising to see that the optimization resulted in using less objects (4) compared to the baseline (8). Would the use of less information not result in a degradation of the performance?
  10. In Section 4.4, an autoencoder on images is described. How are the images constructed from the particles? Assuming that one coordinate corresponds to the azimuthal angle, is there any special treatment to account for the rotational symmetry?
  11. In Section 4.7., is the kernel bandwidth chosen to reflect the resolutions of the objects or is some other criterion used?
  12. In Section 4.12., has the possibility of mode collapse been studied? It would be worth highlighting if an optimization (e.g. of the network architecture or the loss function) has been performed to explicitly mitigate mode collapse.
  13. In section 5.2., consider to clarify that small mass splittings typically means less energetic objects/events and hence the anomalous signal is not in the tails.
  14. In general, it might be interesting to present a pT or HT distribution in Section 5 that demonstrates for one method, how well a signal can be identified by splitting a SM+BSM sample in two parts using the anomaly score (ie. events below and above a threshold). This should give the reader a qualitative idea on how much better an anomaly detection method could perform compared to applying a simple selection on pT or HT instead.

  • validity: good
  • significance: good
  • originality: high
  • clarity: high
  • formatting: perfect
  • grammar: good

Author:  Bryan Ostdiek  on 2021-10-08  [id 1826]

(in reply to Report 2 on 2021-07-21)

The referee writes:

There are some typos (e.g. repeated words) and a few suboptimal descriptions (e.g. last 3 sentences in 5.2.2), which should be fixed.

Our response:Thank you, we have addressed these.

The referee writes:

Figure 1 shows the same histograms, which may confuse a reader since it implies that the anomaly score is or could be equivalent to a supervised classifier.

Our response:This was done intentionally to emphasize the similarities in the analysis strategies. However, as to not confuse the reader, we have commented on this in the caption.

The referee writes:

Various phrases seem to be used as synonyms (e.g. signal region <-> signal area; simulated <-> computer-generated; anomaly values <-> anomaly scores; etc.), which can be confusing. It would be best to use a consistent scheme throughout the paper.

Our response:We have made the language more consistent.

The referee writes:

It would be good to give a bit more details on the analysis objects: i.e. from what objects are jets clustered (e.g. calorimeter cells)?; how are b-jets identified? are leptons/photons required to be isolated? how are ETmiss and its angle defined?

Our response:We have added a link to the Delphes card which was used for the simulation. The b-jet identification is based on ATL-PHYS-PUB-2015-022 and a citation has been added.

The referee writes:

How have the investigated signal scenarios been chosen, e.g. are these already searched for, and if yes point out if the chosen signals have not yet been excluded in direct searches.

Our response: The signal scenarios were chosen to be near the current exclusion limits. While it is possible to choose even larger masses, these are actually easier to find with anomaly detection techniques (or traditional methods), because they look so different from the standard model. The reason that they are not found yet then is because of their small cross-sections. Instead of doing this, we wanted to see how the techniques could work in the signals look closer to the standard model background, even if the cross-section would be too large. We do not use the cross-section in our metrics.

The referee writes:

The chosen channels lack a bit of motivation; e.g. what kind of models can be targeted in each channel?

Our response: Channel 1 looks for hadronic activity with lots of missing energy. This is good for mono-jet type signatures of dark matter as well as any of the colored SUSY signals. Both of the channel 2 options reduce the background by requiring leptons, which then are more sensitive to signals which have an electroweak charge (such as the charginos and neutralinos). Channel 3 is targeted to be more inclusive and catches most of the signals except for the softer electroweak signals. We have commented on this in the data generation section now.

The referee writes:

The shown distributions in Fig. 9 suggests that outliers may always be found in the tail of a distribution. Such events can be easily found as well through a series of selections as in traditional searches. It might instead be worth highlighting that most methods described in the paper could also detect outliers that fall within "voids" in n-dimensions.

Our response:Thank you, we have highlighted this fact.

The referee writes:

A brief discussion should be added to the paper on how to deal with detector noise that might be identified as anomalous as well and can occur at a higher rate than new physics.

Our response: Unlike supervised algorithms, unsupervised algorithms can be trained directly on data, e.g., on a statistically representing fraction of the whole dataset. By doing so, the algorithms would learn detector-related effects as part of the standard representation of the event, making the anomaly detection more robust. We have commented on this on page 5.

The referee writes:

The autoencoder described in Section 4.1 performs better in Channel 2a after an optimization of the hyperparameters. It is however surprising to see that the optimization resulted in using fewer objects (4) compared to the baseline (8). Would the use of less information not result in a degradation of the performance?

Our response: One of the many challenges of anomaly detection with autoencoders is the need to get relatively good reconstruction for the background while simultaneously yielding poor reconstruction for anomalous events. The reason that this section used a re-optimization for Channel 2a was due to the very small training size. With this small size, the network performance was too similar for both the signal and background events when using 8 objects instead of 4. The smaller size also allowed for no zero-padding as all events had at least 4 objects in Channel 2a.

The referee writes:

In Section 4.4, an autoencoder on images is described. How are the images constructed from the particles? Assuming that one coordinate corresponds to the azimuthal angle, is there any special treatment to account for the rotational symmetry?

Our response: The image is not an image in the traditional “jet images” approach. The pixels do not correspond to the eta-phi plane, but instead are a 2D array with dimensions 4 * number_of_particles. For each particle, [E, pT, eta, phi] is the pixel values for the row. As this was unclear, we have updated the text to reflect this. With this, there is not a rotational symmetry of the “image” itself.

The referee writes:

In Section 4.7., is the kernel bandwidth chosen to reflect the resolutions of the objects, or is some other criterion used?

Our response: A data-based maximum likelihood criterium is used, so it should in principle automatically reflect the resolution of the objects AFTER dimensionality reduction. We have commented on this in the section.

The referee writes:

In Section 4.12., has the possibility of mode collapse been studied? It would be worth highlighting if an optimization (e.g. of the network architecture or the loss function) has been performed to explicitly mitigate mode collapse.

Our response: The ALAD GAN is based on BiGANs which inherits implicit regularization, mode coverage, and robustness against mode collapse. We have added this into the section.

The referee writes:

In section 5.2., consider clarifying that small mass splittings typically means less energetic objects/events, and hence the anomalous signal is not in the tails.

Our response: Thank you, we have commented on this on page 38.

The referee writes:

In general, it might be interesting to present a pT or HT distribution in Section 5 that demonstrates for one method, how well a signal can be identified by splitting an SM+BSM sample into two parts using the anomaly score (ie. events below and above a threshold). This should give the reader a qualitative idea of how much better an anomaly detection method could perform compared to applying a simple selection on pT or HT instead.

Our response: We are planning on a follow-up study with more methods and agree that a simple selection would be worthwhile to include. We invite the referee to participate in the study.

---

## Round 1 · Referee Report · Anonymous (Referee 3) · 2021-8-3

Strengths

1- This paper describes different methods to search for new physics, in context of the data collected at the LHC, using a model independent approach. 2- The different methods and algorithms described and studied are based on novel machine learning techniques and a comparison of the effectiveness of the different approaches is presented. 3- The paper uses multiple methods to compare the different algorithms studied, providing an overview of how the different approaches perform under the separate criteria used for evaluation. This provides greater insight into the effectiveness of the different approaches studied. 4- The paper also provides scope for future development and participation for readers by making the code and datasets used available for further studies.

Weaknesses

1- In this paper, the reasons why certain methods perform better than others are not explored in detail, neither are reasons for different performance in different datasets explained - these are left for further study. 2- The likelihood of the algorithms to identify a signal where there is none, due to trials factor or overtraining is not discussed explicitly. 3- The paper does not discuss how experimental effects might affect analyses conducted on real experimental data.

Report

This paper is suitable for publication as it provides a detailed overview of novel methods based on machine learning algorithms to search for new physics in experimental data at the LHC using a model independent approach. A comparison of the effectiveness of the different algorithms is also provided along with information that would provide access to the code and the datasets used for further studies that might be useful for interested readers. The paper is well written and provides adequate description and references for the different algorithms studied. Suggestions for improvements are mentioned as the requested changes.

Requested changes

1- Section 2.1 : Please provide more details of the analysis selection on the objects that are used, for instance it is mentioned that electrons that fall in the crack regions are vetoed, without describing the details of that selection and if that also applies for photons. Further, “a modified version of the ATLAS detector card” is mentioned, can there be a reference or a link provided for this?

2- Section 2.1 : A list of BSM models that are used in this analysis are provided. It would be useful if the paper contained a comment on the choice of the BSM models considered and the motivation for choosing the ones considered, particularly regarding the diversity (or not) in the signatures and the final states they would fall in.

3- Section 2.1 : Regarding the list of channels probed - If possible, it is suggested to add the motivation for this choice of channels. This is considering that the analysis techniques rely on simulation for training, and the reliability of the simulation to describe experimental data is very different in the different channels for standard LHC based analyses. Further, it would be useful to give a brief description of the statistics in each of the channels, particularly since this is discussed later, for example in Section 4.1.

4- Footnote #7 on page 7 mentions that event weights are not used, while in equation 2.6 and surrounding text contains a description of the event weights. This is confusing to the reader and should be clarified.

5- Table 2 mentions single_higgs as well as ttbarHiggs, it is not specified which production modes the single_higgs consists of. Could there be a double counting?

6- Section 4: for some of the methods, the input variables used for the training are explicitly mentioned while they are not for some other methods, it would be more complete to provide this information for each of the methods employed.

7- Section 4.1, paragraph 3 : “leading 8 objects are kept” : please specify if these are leading in pT or E.

8- Section 5.2, paragraph 3 : Since it is mentioned that having less data affects the performance of the algorithms, it would be useful to note if the authors have found there to be a threshold for the amount of data required for training adequately - this is likely to be method dependent.

9- Section 5.3 : A brief description of the blinded dataset, without going into all the details, might be useful to understand the differences.

10- Section 5.3 : Has any study been made on how likely it is for the algorithms to report a high anomaly score for selection regions where there is no signal in a different dataset, that is the output suggests a signal where there is none. This could be due to overtraining or even due to the trials factor. It would be useful to add a discussion to the paper to report if such tests were performed.

Editorial 11- Section 1, Paragraph 1 : “LHC data were initially analysed for various experimental signatures” : It is not clear what is implied by “initially” since such search analyses continue to be performed. 12- Section 1 : A punctuation, such as a colon, after the titles of the different parts that are in bold (eg first few words in Paragraph 1, 4, 9 13) might be suitable. 13- Section 1, Paragraph 6 : LHC searches for new physics -> Searches for new physics at the LHC 14- Page 15, last paragraph : perhaps the full form of GANs should be mentioned in keeping with the description of the other algorithms in the paragraph.

  • validity: high
  • significance: good
  • originality: high
  • clarity: high
  • formatting: excellent
  • grammar: good

Author:  Bryan Ostdiek  on 2021-10-08  [id 1827]

(in reply to Report 3 on 2021-08-03)

The referee writes:

In this paper, the reasons why certain methods perform better than others are not explored in detail, neither are the reasons for different performances in different datasets explained - these are left for further study.

Our response:We agree. The fact that the fixed target methods performed so well was unexpected. Exploring the reasons for their good performance is new ongoing work.

The referee writes:

The likelihood of the algorithms to identify a signal where there is none, due to trials factor or overtraining is not discussed explicitly.

Our response: While there could be a large trial factor if using many unsupervised methods, or if predefined signal regions are not used, we believe these analyses can also be done with small trial factors. One a preselection is done to define the channel 1. Train an algorithm defining an anomaly score 2. Define Signal Regions with the anomaly score (with 10-3 etc background events) 3. Define control regions as in many other searches 4. Predict the background with a background-only fit 5. Determine a p-value/significance with the data and the background prediction in your signal region In this way, there are no trial factors (or only as many trial factors as there are signal regions that are defined). This is now discussed in Section 1.

The referee writes:

The paper does not discuss how experimental effects might affect analyses conducted on real experimental data.

Our response: We believe that the strategy outlined above should work well for real experimental data. The other option that unsupervised learning allows for is to train directly on the real data itself, assuming that it is background-dominated. Then the most anomalous events could be studied later.

The referee writes:

Section 2.1: Please provide more details of the analysis selection on the objects that are used, for instance, it is mentioned that electrons that fall in the crack regions are vetoed, without describing the details of that selection and if that also applies for photons. Further, “a modified version of the ATLAS detector card” is mentioned, can there be a reference or a link provided for this?

Our response: The crack regions are identified in the second bullet of the section. We have updated the document to include a link to the Delphes detector card used.

The referee writes:

Section 2.1: A list of BSM models that are used in this analysis is provided. It would be useful if the paper contained a comment on the choice of the BSM models considered and the motivation for choosing the ones considered, particularly regarding the diversity (or not) in the signatures and the final states they would fall in.

Our response: As the Dark Machines groups, we are interested in exploring matters of dark matter with machine learning. With that in mind, all of the models explored had some component of dark matter escaping the detector. While these are certainly not all of the interesting BSM models that could be discovered, it is what led to the model selection.

The referee writes:

Section 2.1: Regarding the list of channels probed - If possible, it is suggested to add the motivation for this choice of channels. This is considering that the analysis techniques rely on simulation for training, and the reliability of the simulation to describe experimental data is very different in the different channels for standard LHC-based analyses. Further, it would be useful to give a brief description of the statistics in each of the channels, particularly since this is discussed later, for example in Section 4.1.

Our response: Thank you for this suggestion. We have added in some basic motivation for the channels as well as the statistics for the training. Channel 1 looks for hadronic activity with lots of missing energy. This is good for mono-jet type signatures of dark matter as well as any of the colored SUSY signals. Both of the channel 2 options reduce the background by requiring leptons, which are then more sensitive to signals which have an electroweak charge (such as the charginos and neutralinos). Channel 3 is targeted to be more inclusive and catches most of the signals except for the softer electroweak signals. The size of the datasets are 214,000 SM events for Channel 1; 20,000 SM events for Channel 2a; 340,000 SM events for Channel 2b; and 8,500,000 for Channel 3.

The referee writes:

Footnote #7 on page 7 mentions that event weights are not used, while equation 2.6 and the surrounding text contain a description of the event weights. This is confusing to the reader and should be clarified.

Our response: The event weights are not used in this study. We mention it because it is available in the public data and could be useful for future studies.

The referee writes:

Table 2 mentions single_higgs as well as ttbarHiggs, it is not specified which production modes the single_higgs consists of. Could there be double counting?

Our response:The ttbarHiggs was generated with the MadGraph process: * model SM * generate p p > t t~ h @0 * add process p p > t t~ h j @1 The Single Higgs process is through gluon fusion and done in MadGraph with a model with an effective gluon-gluon-Higgs coupling: * generate p p > h @0 * add process p p > h j @1 * add process p p > h j j @2 [also includes VBF] Jets are defined to not include the top quark, so these should be independent of each other with no double counting. Other details can be found at https://github.com/melli1992/unsupervised_darkmachines.

The referee writes:

Section 4: for some of the methods, the input variables used for the training are explicitly mentioned while they are not for some other methods, it would be more complete to provide this information for each of the methods employed.

Our response: That you for pointing this out. We have updated all of the methods to describe the input variables.

The referee writes:

Section 4.1, paragraph 3: “leading 8 objects are kept”: please specify if these are leading in pT or E.

Our response: Leading in pT. We have updated the text to reflect this.

The referee writes:

Section 5.2, paragraph 3: Since it is mentioned that having fewer data affects the performance of the algorithms, it would be useful to note if the authors have found there to be a threshold for the amount of data required for training adequately - this is likely to be method dependent.

Our response: Unfortunately, this is quite model-dependent. However, it does seem like there are possibly two thresholds, in general. The first is that there needs to be enough data to get a good model, and channel 2a is too small for most methods. The other threshold is when there gets to be so much data that training becomes inefficient or too slow.

The referee writes:

Section 5.3: A brief description of the blinded dataset, without going into all the details, might be useful to understand the differences.

Our response: We prefer to keep the dataset completely blind so that it can be used in the future without bias.

The referee writes:

Section 5.3: Has any study been made on how likely it is for the algorithms to report a high anomaly score for selection regions where there is no signal in a different dataset, that is the output suggests a signal where there is none. This could be due to overtraining or even due to the trials factor. It would be useful to add a discussion to the paper to report if such tests were performed.

Our response: No explicit test like this was done. The methods were trained using 90% of the SM backgrounds and then the anomaly score cuts were implemented on the remaining 10%. As these are statistically independent we do not expect there to be large differences between training sets. Differences in simulations (or experimental results) could have an impact, but this is left for future research.

The referee writes:

Section 1, Paragraph 1: “LHC data were initially analyzed for various experimental signatures”: It is not clear what is implied by “initially” since such search analyses continue to be performed.

Our response: We have fixed this by removing the past tense.

The referee writes:

Section 1: A punctuation, such as a colon, after the titles of the different parts that are in bold (eg first few words in Paragraph 1, 4, 9 13) might be suitable.

Our response: We have added a colon after these.

The referee writes:

Section 1, Paragraph 6: LHC searches for new physics -> Searches for new physics at the LHC

Our response: This has been fixed.

The referee writes:

Page 15, last paragraph: perhaps the full form of GANs should be mentioned in keeping with the description of the other algorithms in the paragraph.

Our response: We have changed GAN to “generative adversarial network.”

---

## Round 2 · Referee Report · Anonymous · 2021-11-18

Strengths

1- This paper describes different methods to search for new physics, in context of the data collected at the LHC, using a model independent approach.
2- The different methods and algorithms described and studied are based on novel machine learning techniques and a comparison of the effectiveness of the different approaches is presented.
3- The paper uses multiple methods to compare the different algorithms studied, providing an overview of how the different approaches perform under the separate criteria used for evaluation. This provides greater insight into the effectiveness of the different approaches studied.
4- The paper also provides scope for future development and participation for readers by making the code and datasets used available for further studies.

Weaknesses

1- In this paper, the reasons why certain methods perform better than others are not explored in detail, neither are the reasons for different performances in different datasets explained - these are left for further study.

Report

This paper is suitable for publication as it provides a detailed overview of novel methods based on machine learning algorithms to search for new physics in experimental data at the LHC using a model independent approach. A comparison of the effectiveness of the different algorithms is also provided along with information that would provide access to the code and the datasets used for further studies that might be useful for interested readers.

The authors have addressed the requested changes submitted in the previous round of review, some minor follow up is suggested in the requested changes to improve the understanding of readers that would not have access to the authors' reply.

Requested changes

1- The authors have mentioned in their response that - "As the Dark Machines groups, we are interested in exploring matters of dark matter with machine learning. With that in mind, all of the models explored had some component of dark matter escaping the detector. While these are certainly not all of the interesting BSM models that could be discovered, it is what led to the model selection." -> suggest to indicate this motivation of choice of BSM models in the paper explicitly.

2. While addressing the suggestion on including the size of the dataset, the numbers mentioned for Channel 1 in their reply ("size of the datasets are 214,000 SM events for Channel 1 ") and that stated in the text of the paper is different. The authors can resolve this minor ambiguity.

3. In the meanwhile between the submission of the first draft of the paper for review, an updated version of Reference 16 (CMS Collaboration, MUSiC ... ) has been published : Eur. Phys. J. C 81, 629 (2021); this reference can be updated.

  • validity: high
  • significance: high
  • originality: high
  • clarity: good
  • formatting: excellent
  • grammar: good

Author:  Bryan Ostdiek  on 2021-12-10  [id 2021]

(in reply to Report 1 on 2021-11-18)

Thank you for the final suggestions. We have fixed these in the new version.

---

## Round 2 · Referee Report · Anonymous · 2021-11-19

Report

Thanks for the revised document. I see that most of my comments have been addressed sufficiently. I have no further comments.

---

## Editorial Decision

published